# Multiple evolutionary origins and losses of tooth complexity in squamates

Fabien Lafuma [1✉], Ian J. Corfe [1,2✉], Julien Clavel[3,4] & Nicolas Di-Poï [1✉]

Teeth act as tools for acquiring and processing food, thus holding a prominent role in vertebrate evolution. In mammals, dental-dietary adaptations rely on tooth complexity variations controlled by cusp number and pattern. Complexity increase through cusp addition has dominated the diversification of mammals. However, studies of Mammalia alone cannot reveal patterns of tooth complexity conserved throughout vertebrate evolution. Here, we use morphometric and phylogenetic comparative methods across fossil and extant squamates to show they also repeatedly evolved increasingly complex teeth, but with more flexibility than mammals. Since the Late Jurassic, multiple-cusped teeth evolved over 20 times independently from a single-cusped common ancestor. Squamates frequently lost cusps and evolved varied multiple-cusped morphologies at heterogeneous rates. Tooth complexity evolved in correlation with changes in plant consumption, resulting in several major increases in speciation. Complex teeth played a critical role in vertebrate evolution outside Mammalia, with squamates exemplifying a more labile system of dental-dietary evolution.

[1] Institute of Biotechnology, Helsinki Institute of Life Science, University of Helsinki, FI-00014 Helsinki, Finland. [2] Geological Survey of Finland, FI-02150 Espoo, Finland. [3] Department of Life Sciences, The Natural History Museum, London SW7 5BD, UK. [4] Univ. Lyon, Université Claude Bernard Lyon 1, CNRS, ENTPE, UMR 5023 LEHNA, F-69622 Villeurbanne, France. ✉email: fabien.lafuma@gmail.com; ian.corfe@gtk.fi; nicolas.di-poi@helsinki.fi

As organs directly interacting with the environment, teeth are central to the acquisition and processing of food, determine the achievable dietary range of vertebrates[1], and their shapes are subject to intense natural selective pressures[2,3]. Simple conical to bladed teeth generally identify faunivorous vertebrates, while higher dental complexity—typically a result of more numerous cusps—enables the reduction of fibrous plant tissue and is crucial to the feeding apparatus in many herbivores[2,4,5]. Evidence of such dental-dietary adaptations dates back to the first herbivorous tetrapods in the Palaeozoic, about 300 million years ago (Ma)[4]. Plant consumers with highly complex teeth have subsequently emerged repeatedly within early synapsids[4], crocodyliforms[6], dinosaurs[7–9], and stem and crown mammals[10–13]. Since the earliest tetrapods had simple, single-cusped teeth[2], such examples highlight repeated, independent increases of phenotypic complexity throughout evolution[14]. Many such linked increases in tooth complexity and plant consumption have been hypothesised to be key to adaptive radiations[11,13], though such links have rarely been formally tested. It is also unclear whether the known differences in tooth development between tetrapod clades might result in differences in the evolutionary patterns of convergent functional adaptations[15,16].

To understand the repeated origin of dental-dietary adaptations and their role in vertebrate evolution, we investigated tooth complexity evolution in squamates ("lizards" and snakes), the largest tetrapod radiation. Squamates can have simple teeth in homodont dentitions or complex, multicuspid teeth in heterodont dentitions[17], and squamate ecology spans a broad range of past and present niches. Squamates express dental marker genes broadly conserved across vertebrates[15], with varying patterns of localisation and expression compared to mammals, and structures at least partially homologous to mammalian enamel knots (non-proliferative signalling centres of ectodermal cells) determine tooth shape in some squamate clades[16,18,19]. In mammals—the most commonly studied dental system—novel morphologies arise from developmental changes in tooth morphogenesis[20]. Epithelial signalling centres—the enamel knots—control tooth crown morphogenesis[21], including cusp number and position and ultimately tooth complexity, by expressing genes of widely conserved signalling pathways[15,22]. Experimental data show most changes in these pathways result in tooth complexity reduction, or complete loss of teeth[22]. Yet, increasing tooth complexity largely dominates the evolutionary history of mammals[2,10–13], and it remains unknown whether similar patterns of tooth complexity underlie all tetrapod evolution or are the specific results of mammalian dental development and history.

Here, we assemble and analyse a dataset of cusp number and diet data for 545 squamate species spanning all the living and extinct diversity of the group since its estimated origin in the Permian[23], including tooth morphometric data for 75 species representing all past and present clades we find to have evolved complex teeth. First, we examine the extent of squamate dental diversity and its distribution throughout the group and test how cusp number and tooth shape relate to squamate diets. We then use phylogenetic comparative methods to investigate the patterns of evolution of tooth complexity and plant consumption and address their connection through evolutionary history. Lastly, we fit two different model frameworks to determine whether dental complexity and diet evolution drove squamate diversification. Our findings show that squamate tooth complexity evolved and was lost numerous times in correlation with the evolution of plant-based diets, with the Late Cretaceous representing a critical time for both traits. The Late Cretaceous was also a crucial period for squamate diversification dynamics, and we find that higher tooth complexity and plant consumption led to higher speciation rates.

## Results

**Dental-dietary adaptations to plant consumption.** Among the 545 fossil and extant squamate species examined, we identified species with multicuspid teeth in 29 of 100 recognised squamate families (Fig. 1a, Supplementary Figs. 1–4, Supplementary Data 1–3). Within extant "lizards", we found multicuspid species in almost 56% of families (24/43), with three-cusped teeth being most common (67% of multicuspid species). While lacking entirely in mostly predatory clades (dibamids, geckos, snakes), multicuspidness dominates Iguania and Lacertoidea, the two most prominent groups of plant-eating squamates, totalling 72% of all extant omnivores and herbivores (Fig. 1a and Supplementary Data 3). A Kruskal–Wallis test and post hoc pairwise Wilcoxon–Mann–Whitney tests show squamate dietary guilds differ statistically in tooth complexity, with the proportion of multicuspid species and cusp number successively increasing along a plant consumption gradient, from carnivores to insectivores, omnivores, and herbivores ($p$-value $< 0.001$; Fig. 1b and Supplementary Table 1). We quantified tooth outline shape in a subset of taxa spanning all major multicuspid groups with two-dimensional semi-landmarks (Supplementary Data 4). Along principal component 1 (PC1) of the multicuspid tooth morphospace, tooth protrusion is responsible for the greatest extent of shape variation (80.82%), from low-protruding teeth at negative scores, to high-protruding teeth at positive scores (Fig. 1c). Principal component 2 (PC2, 8.08% of total variance) reflects changes in top cusp angle (i.e., apical flaring), which increases when going from negative to positive scores (i.e., conical to fleur-de-lis-like teeth, Fig. 1c). In contrast to the relatively similar teeth of insectivores and omnivores, the teeth of herbivores occupy a distinct morphospace region of more protruding morphologies with a wider top cusp angle (Fig. 1c). A regularised phylogenetic multivariate analysis of variance (MANOVA) on principal component scores confirms statistically significant differences between diets overall ($p$-value $= 8.0e-04$; Fig. 1c) with negligible phylogenetic signal in the model's residuals (Pagel's $\lambda = 0.03$). Herbivore teeth significantly differ from both the insectivorous and omnivorous morphospace regions (Fig. 1c and Supplementary Table 2), similarly to observations from mammals and saurians[5,17]. Conversely, insectivores and omnivores share a largely overlapping morphospace, which is maintained even in discriminant analysis (DFA; Supplementary Fig. 5).

**Patterns in tooth complexity and diet evolution.** Using Maximum Likelihood reconstructions of ancestral character states across our squamate phylogeny (Fig. 2, Supplementary Figs. 6 and 7, Supplementary Tables 3 and 4), we found dental-dietary adaptations to plant consumption repeatedly evolved, arising from the convergent evolution of multicuspidness. Since the Late Jurassic, six major clades and 18 isolated lineages independently evolved multicuspid teeth from a unicuspid ancestral morphology, mostly through single-cusp addition events, for a total of 24 independent originations of complex teeth (see Fig. 2 and Supplementary Fig. 6 for the names and positions of major clades 1–6). Similar numbers —10 major clades, 13 isolated lineages—show independent origins of plant consumption from carnivorous or insectivorous ancestors (see Fig. 2 and Supplementary Fig. 6 for the names and positions of major clades 1′–10′). Across the tree, most lineages and terminal taxa are unicuspid insectivores retaining the reconstructed ancestral squamate condition. However, of 102 lineages showing cusp number or plant consumption increases, 42 (41%) of increases are along the same phylogenetic path as an increase in the other character (see Methods; Supplementary Table 5).

Although both tooth complexity and plant consumption first originated relatively early (in the Late Jurassic and Early

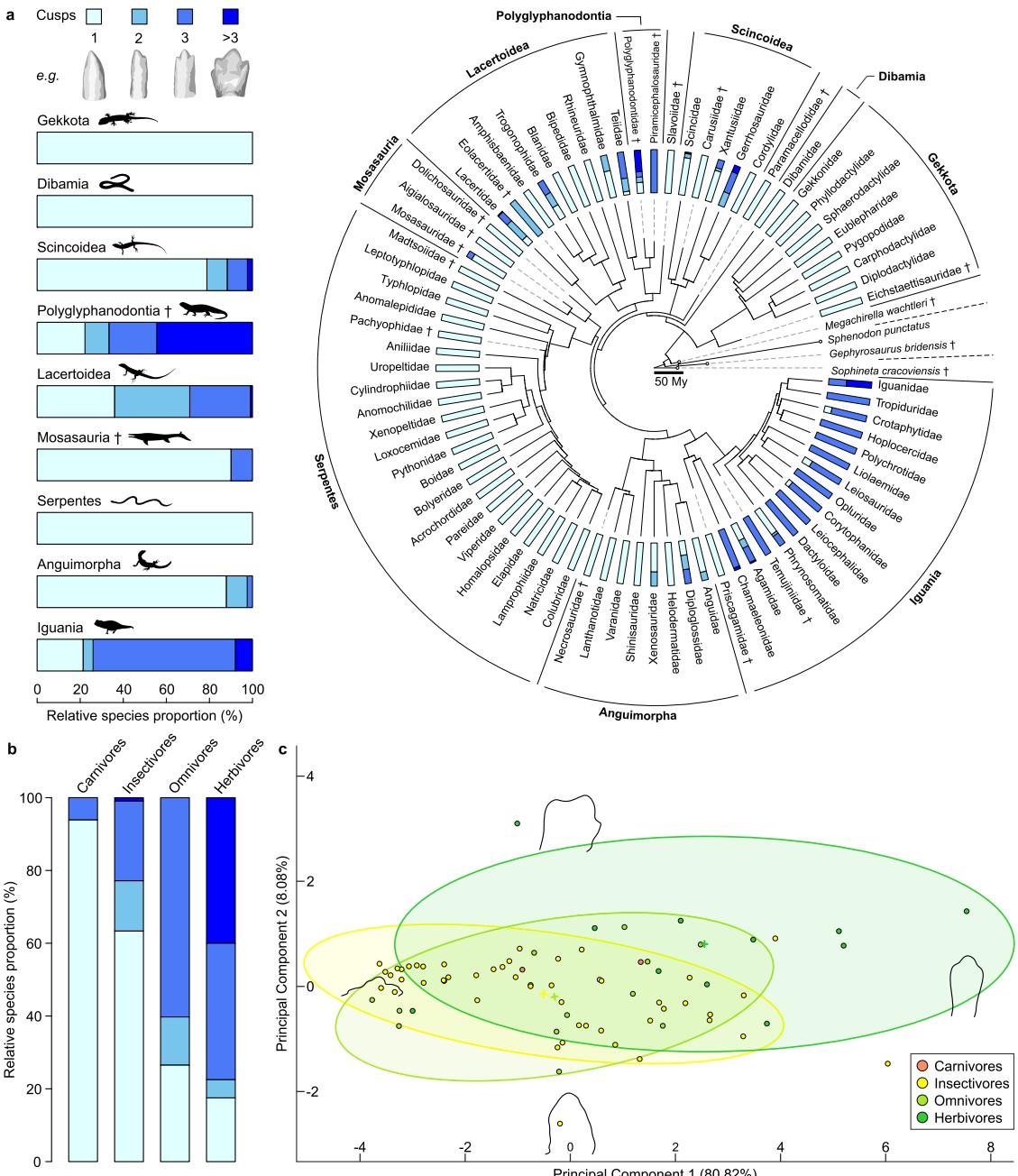

**Fig. 1 The diversity of squamate dental morphologies correlates with a gradient of plant consumption. a** Relative proportions (%) of tooth complexity levels in all known squamate suborders/super-families (left) and 76 families (right) based on cusp number data for 545 living and extinct species (including the most ancient known squamate *Megachirella wachtleri*), two rhynchocephalians, and the stem lepidosaurian *Sophineta cracoviensis*, with example teeth for each complexity level redrawn from microCT-scan data (not to scale). **b** Relative proportions (%) of tooth complexity levels in 545 squamates sorted by diet. **c** Discrete cosine transform analysis of multicuspid tooth labial view profiles from 75 extant and fossil squamate species, with 95% confidence ellipses for insectivorous, omnivorous, and herbivorous morphologies. Theoretical tooth profiles at the extreme positive and negative values of each axis reconstructed from the first 21 harmonic coefficients. Scalebar = 50 million years (My). Dagger = extinct taxon. Silhouettes: Phylopic (http://phylopic.org) courtesy of T. Michael Keesey (used without modification, CC0 1.0 and CC-BY 3.0 https://creativecommons.org/licenses/by/3.0/), David Orr (CC0 1.0), Iain Reid (used without modification, CC-BY 3.0 https://creativecommons.org/licenses/by/3.0/), Alex Slavenko (CC0 1.0), and Steven Traver (CC0 1.0), and F.L. after Darren Naish (used with permission) and Ted M. Townsend (silhouette drawn from photograph, CC-BY 2.5 https://creativecommons.org/licenses/by/2.5/deed.en); see Methods for full license information. Source data are provided as a Source Data file.

Cretaceous, respectively), they remained marginal for dozens of millions of years (Fig. 3a, c). It is only during the Cretaceous Terrestrial Revolution (KTR, ~125–80 Ma) and, specifically, the Late Cretaceous, that the relative abundance and disparity of multiple-cusped teeth and plant-based diets started increasing.

The highest cusp numbers and true herbivorous diets became increasingly prevalent only after the Cretaceous–Paleogene (K–Pg) mass extinction. Importantly, squamate dental evolution was labile and included repeated reversals towards lower tooth complexity. Both complexity and diet changed similarly through

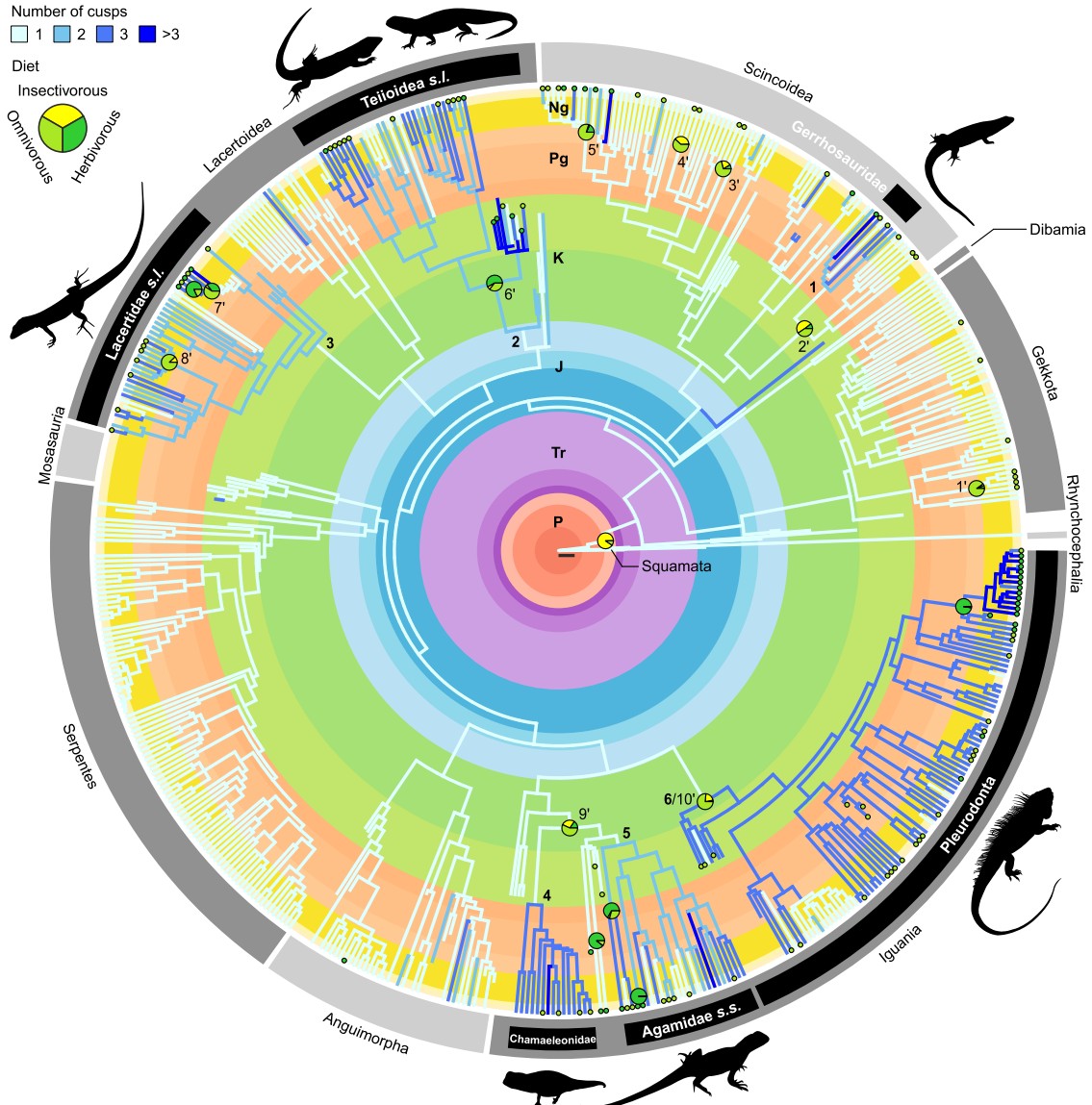

**Fig. 2 Multiple independent acquisitions of multicuspid teeth and plant consumption are found across squamate phylogeny.** Known and maximum likelihood ancestral state reconstructions of the number of cusps (branch colours) and diet (node pie charts and branch tip small circles) in squamates. Pie charts indicate the most ancient nodes with >50% combined relative likelihood for omnivorous and herbivorous diets; also shown are the first nodes with >50% relative likelihood for herbivory within already omnivorous clades. Branch tip circles indicate omnivorous/herbivorous species. Six major clades show independent originations of multicuspid teeth: 1: Gerrhosauridae (node 616; see Supplementary Fig. 1). 2: Teiioidea + Polyglyphanodontia (informally Teiioidea *sensu lato*; node 686). 3: total group Lacertidae (informally Lacertidae *sensu lato*; node 740). 4: Chamaeleonidae (node 930). 5: non-Uromastycinae agamids (informally Agamidae *sensu stricto*; node 949). 6: total group Pleurodonta (node 971). Independent originations of plant consumption (isolated terminal branches not included): 1′: unnamed clade including the most recent common ancestor (MRCA) to *Correlophus ciliatus* and *Rhacodactylus auriculatus* and all its descendants (node 563). 2′: Cordyloidea (node 609). 3′: unnamed clade including the MRCA to *Eumeces schneideri* and *Scincus scincus* and all its descendants (node 652). 4′: *Chalcides* (node 657). 5′: Egerniinae (node 674). 6′: unnamed clade including the MRCA to *Polyglyphanodon sternbergi* and *Teius teyou* and all its descendants (node 688). 7′: *Gallotia* (node 750). 8′: *Podarcis* (node 765). 9′: crown Acrodonta (node 929). 10′: total group Pleurodonta (node 971). P: Permian. Tr: Triassic. J: Jurassic. K: Cretaceous. Pg: Paleogene. Ng: Neogene. Scalebar = 10 million years. Silhouettes: Phylopic (http://phylopic.org) courtesy of T. Michael Keesey (from a photograph by Frank Glaw, Jörn Köhler, Ted M. Townsend & Miguel Vences, used without modification, CC-BY 3.0 https://creativecommons.org/licenses/by/3.0/), Michael Scroggie (CC0 1.0), Alex Slavenko (CC0 1.0), and Jack Meyer Wood (CC0 1.0), and F.L. after Dick Culbert (silhouette drawn from photograph, CC-BY 2.0 https://creativecommons.org/licenses/by/2.0/deed.en), Scott Robert Ladd (silhouette drawn from photograph, CC-BY 3.0 https://creativecommons.org/licenses/by/3.0/), and Darren Naish (used with permission); see Methods for full license information.

much of squamate evolutionary history, though there were more changes in diet than complexity (115 vs. 92 lineages), and reversals in diet were more common than for complexity (56% vs. 44%) (Fig. 3b, d and Supplementary Fig. 6). Such flexibility is reflected in the reconstructed transition rates underlying our models of evolution for tooth complexity and diet, where higher relative rates characterise decreases in cusp number and plant consumption compared to increases (Supplementary Fig. 7 and Supplementary Tables 3 and 4). Moreover, 38% of inferred complexity decreases were due to the simultaneous loss of two cusps or more, while multiple-cusp addition events were half (20%) as frequent. We identify two lineages (genera *Gallotia* and

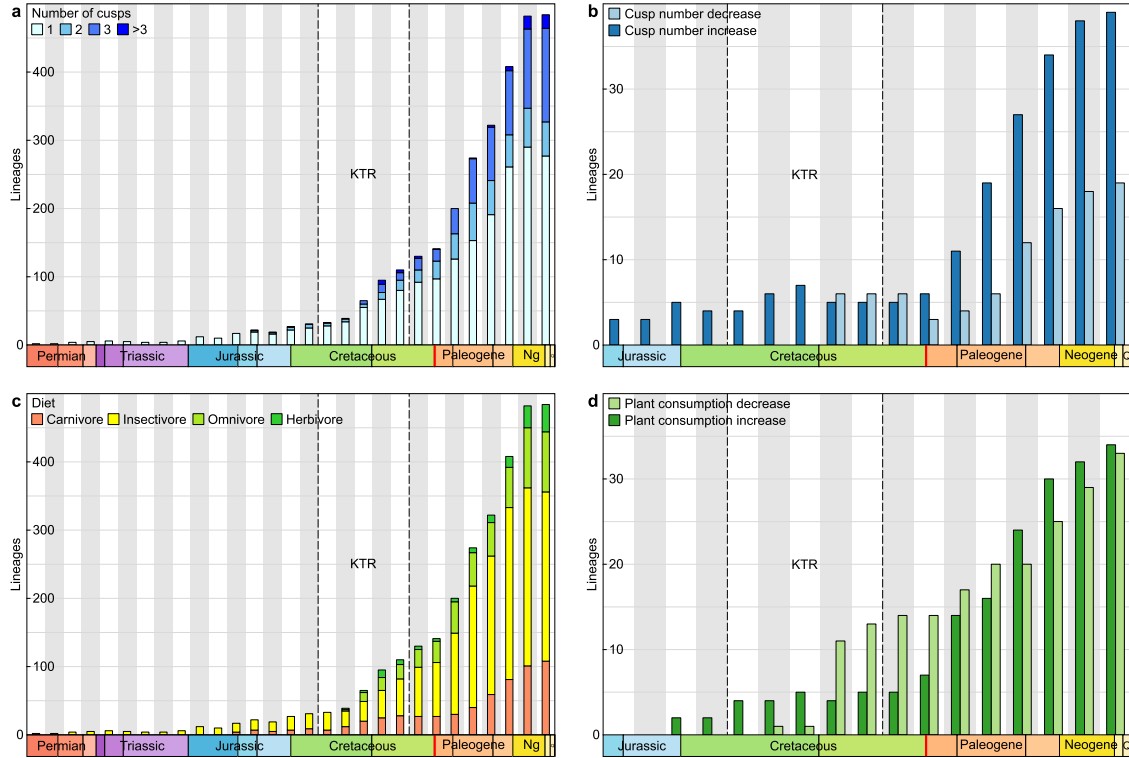

**Fig. 3 Dynamics of squamate tooth complexity and plant consumption evolution.** Squamate lineages sorted by cusp number (**a**, single-cusped: $n = 651$, two-cusped: $n = 145$, three-cusped: $n = 238$, more than three cusps: $n = 33$) and diet (**c**, carnivore: $n = 246$, insectivore: $n = 556$, omnivore: $n = 203$, herbivore: $n = 62$) per 10 million year-time bins based on maximum likelihood ancestral state reconstructions. Lineages showing increasing ($n = 61$) or decreasing ($n = 31$) tooth complexity (**b**) and increasing ($n = 51$) or decreasing ($n = 64$) plant consumption (**d**) per 10 million year-time bins. KTR: Cretaceous Terrestrial Revolution (~125–80 Ma). Q: Quaternary. Decreases in both cusp number and plant consumption proportion first outnumber increases during the Cretaceous Terrestrial Revolution (KTR), while the Cretaceous–Paleogene boundary (in red) shows the change towards the Cenozoic pattern of approximately twice as many cusp increases as decreases, and similar numbers of plant consumption increase and decrease from the Paleogene on. Source data are provided as a Source Data file.

*Phrynosoma*) in which multicuspid teeth re-evolved following earlier loss (Fig. 2 and Supplementary Fig. 6). Most often, reversals to lesser cusp numbers followed a decrease in plant consumption (52% of paired events; Supplementary Table 5), likely resulting from the relaxation of selective pressures for plant consumption.

Furthermore, we find the observed dental-dietary patterns derive from the correlated evolution of tooth complexity and plant-based diets under highly variable rates of phenotypic evolution. Our results show strong support for a correlated model of the evolution of multicuspidness and plant consumption, which assumes transition rates in one trait directly depend on character state in the other trait (log Bayes Factor = 21, Supplementary Table 6). Additionally, a model with heterogenous character transition rates throughout the tree better fits the macroevolutionary pattern of each trait than a constant model, with the highest rates observed resulting in a relatively balanced mixture of tooth complexity and diet character states for the clades concerned (e.g., Lacertidae) (Figs. 2, 4a, b and Supplementary Table 7). We also find pronounced support for shifts in the rate of evolution of dental shape outline independent of cusp number among the 75 species with multicuspid teeth examined (log Bayes Factor = 319), with particularly high rates characterising Iguanidae (Supplementary Fig. 8). A multiple-optima Ornstein-Uhlenbeck multivariate model is largely preferred to fit tooth outline data, showing that the evolution of multicuspid tooth shapes can be explained by a dietary-

dependent adaptive process (Supplementary Tables 8–11 and Supplementary Fig. 9).

**Squamate diversification dynamics.** These evolutionary increases in tooth complexity and plant consumption appear to have contributed to the diversification of Squamata. Using models with variable rates of diversification implemented in a Bayesian framework through a reversible jump Markov Chain Monte Carlo algorithm and allowing for the inclusion of fossil taxa (see Methods), we identified 18 events of increased speciation with up to an eight-fold magnitude for the focal group vs. its outgroup, including 13 shifts repeated in at least half our replicates (Fig. 4c and Supplementary Table 12). Five speciation increases—all inferred in half of replicates or more—coincide exactly with increases in tooth complexity (Polyglyphanodontia and a sub-clade of Iguanidae), plant consumption (Egerniinae and *Podarcis*, both towards omnivory) or both (total group Pleurodonta, see Fig. 4c and Supplementary Table 12), and five are just one node away from such increases (Polyglyphanodontia, total group Teiioidea, crown Teiioidea, the previous Iguanidae subclade, and one of its own sub-clades, see Supplementary Table 13). The equivalent results for decreases are two (total group *Gallotia*, in only two replicates, and genus *Phrynosoma*, in all replicates, see Fig. 4c and Supplementary Table 12) and three lineages, respectively (Polyglyphanodontia, crown Pleurodonta, and a subclade of Liolaemidae, see Supplementary Table 13). Furthermore, five of

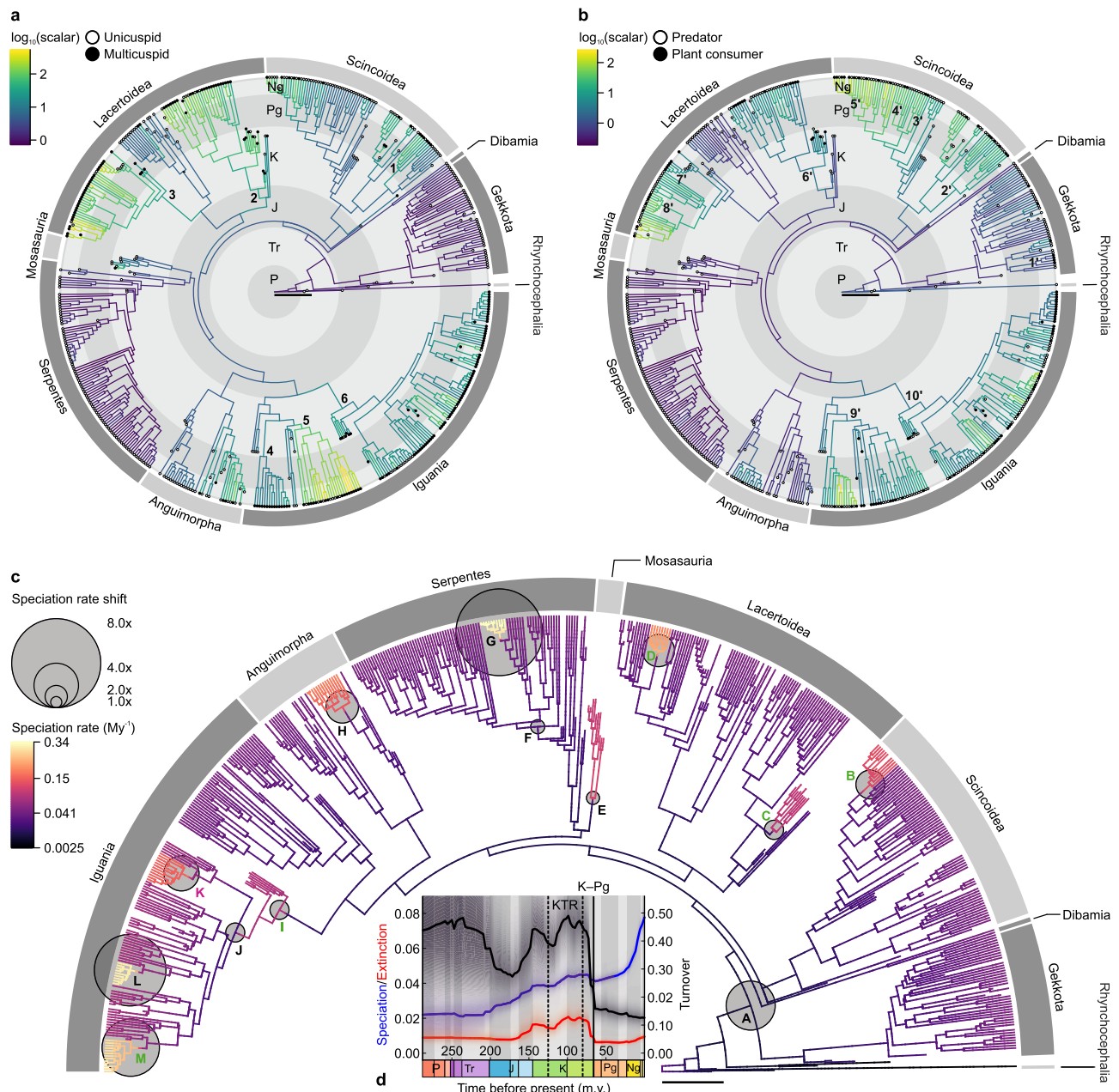

**Fig. 4 Dental-dietary rates of phenotypic evolution and squamate diversification patterns.** Log-transformed averaged rate scalars of the character transition rates of tooth complexity (**a**) and diet (**b**) across squamates. Positive values (i.e., rate scalar > 1) indicate increased relative transition rates. **c** Rates of squamate speciation for one maximum shift credibility configuration (MSC) out of ten similar independent replicates. Thirteen rate shifts (A–M) present in at least five MSC replicates are indicated proportionally to their magnitude compared to the background rate. **d** Mean rates of squamate speciation and extinction (in My⁻¹), and turnover (extinction/speciation) through time; see also Supplementary Fig. 10. Shaded areas: 95% confidence interval. 1: Gerrhosauridae (node 616; see Supplementary Fig. 1). 2: Teiioidea + Polyglyphanodontia (informally Teiioidea *sensu lato*, node 686). 3: total group Lacertidae (informally Lacertidae *sensu lato*, node 740). 4: Chamaeleonidae (node 930). 5: non-Uromastycinae agamids (informally Agamidae *sensu stricto*, node 949). 6: total group Pleurodonta (node 971). 1′: unnamed clade including the most recent common ancestor (MRCA) to *Correlophus ciliatus* and *Rhacodactylus auriculatus* and all its descendants (node 563). 2′: Cordyloidea (node 609). 3′: unnamed clade including the MRCA to *Eumeces schneideri* and *Scincus scincus* and all its descendants (node 652). 4′: *Chalcides* (node 657). 5′: Egerniinae (node 674). 6′: unnamed clade including the MRCA to *Polyglyphanodon sternbergi* and *Teius teyou* and all its descendants (node 688). 7′: *Gallotia* (node 750). 8′: *Podarcis* (node 765). 9′: crown Acrodonta (node 929). 10′: total group Pleurodonta (node 971). A: crown Squamata. B: Egerniinae. C: Polyglyphanodontia. D: *Podarcis*. E: Mosasauria. F: crown Alethinophidia. G: *Chilabothrus*. H: *Varanus*. I: total group Pleurodonta. J: crown Pleurodonta. K: *Phrynosoma*. L: unnamed clade including the most recent common ancestor (MRCA) of *Liolaemus darwinii* and *L. scapularis* and all its descendants. M: unnamed clade including the MRCA of *Ctenosaura quinquecarinata* and *Cyclura cornuta* and all its descendants. Direct correspondences between increased speciation and increased (green) or decreased (magenta) tooth complexity/plant consumption are indicated. P: Permian. Tr: Triassic. J: Jurassic. K: Cretaceous. Pg: Paleogene. Ng: Neogene. Q: Quaternary. KTR: Cretaceous Terrestrial Revolution (~125–80 Ma). K–Pg: Cretaceous–Paleogene extinction event (66 Ma). Scalebars = 50 million years.

the 13 shifts towards increased speciation occurred within a single clade—Pleurodonta—coinciding and following the evolution of three-cusped teeth and omnivory. Notable deviations include Mosasauria, crown Alethinophidia, *Chilabothrus*, and *Varanus* (see Fig. 4c and Supplementary Fig. 12), four clades of predatory squamates with single-cusped teeth that experienced marked increases in speciation rate. Over the whole of Squamata, the mean rate of speciation locally peaked in the Late Cretaceous, towards the end of the KTR. Mean extinction and turnover (extinction/speciation) reached their absolute peak during the same period (Fig. 4d and Supplementary Fig. 10), and a local maximum in species diversity resulted immediately following the end of the KTR, in the Campanian (Supplementary Fig. 11). We further tested the apparent association between diversification shifts and transitions of cusp number and diet using a hidden state trait-dependent model of speciation and extinction. Results from this model (Supplementary Table 14 and Supplementary Fig. 12) suggest each trait (tooth complexity and diet) contributed considerably to the diversification of the group as a whole and particularly of non-ophidian squamates—with rates of speciation and extinction increasing with transitions to multicuspidness or plant-based diets—despite the influence of unobserved factors beyond our study. Combining these results, we propose plant consumption and tooth complexity changes—principally increases—were critical innovations for squamate evolution.

## Discussion

The evolution of tooth complexity in Squamata encompasses multiple independent radiations defined by increasingly complex teeth. This mirrors patterns of mammalian diversification, in which stem mammals show repeated independent evolutions of multicuspid teeth through the Palaeozoic and Mesozoic[4,12,13], the key adaptations of tribosphenic or pseudo-tribosphenic molars separately originated in the Jurassic[24], and quadritubercular molars with a hypocone appeared multiple times in the Cenozoic[11,25,26]. It differs from the mammalian pattern, however, in that the most recent common ancestor (MRCA) of Mammalia was multicuspid[12]. Here, we reconstruct the MRCA of Squamata as unicuspid and infer at least 24 independent acquisitions of multicuspidness in squamate lineages, a number that might increase in future studies including additional extant or fossil specimens. Squamate tooth evolution was also not mainly unidirectional as in mammals, with numerous lineages losing tooth complexity, including reversals to the ancestral unicuspid condition, though never reaching complete loss of teeth. Moreover, tooth complexity at times subsequently re-emerged within lineages that previously underwent such reversals, in opposition to Dollo's law[27]. Despite the lack of a similar large-scale phylogenetic assessment, studies suggest relatively few mammalian lineages experienced reversals towards reduced tooth complexity (including complete tooth loss)[28–30], and even fewer re-evolved cusps once lost[31]. These discrepancies highlight that how often dental complexity evolves is arguably less critical than how complexity emerges in a clade. The single origin of the tribosphenic molar led to therian mammals achieving degrees and patterns of tooth complexity unparalleled across vertebrates and considerable ecological diversification during the Cenozoic. In contrast, by the Late Cretaceous several squamate lineages had evolved multicuspid teeth, even reaching uniquely complex morphologies[32]. Yet, squamates never recovered some of these morphologies and failed to increase dental complexity further following the K–Pg extinction, meaning that most of squamate dental diversity can largely be described by the sole variation of cusp number. Other factors beyond the scope of this study are nevertheless noteworthy. Most notably, squamate multicuspid

morphologies may differ in the accommodation of cusp placement on the crown: most often, along the mesial-distal axis with varying degrees of lateral crown compression or, less commonly, bucco-lingually through an extension of the crown such as in extant Teiidae (e.g., *Teius teyou*[33]) or extinct Polyglyphanodontia (e.g., *Peneteius aquilonius*[32]). Few squamates are distinguished by a degree of occlusion between tooth rows[32,34], which may actively contribute to tooth morphology through dental wear[35]. Morphological heterodonty along the dental row is another important feature of species bearing multicuspid teeth, ranging from an anterio-posterior gradient of multicuspidness to regionalisation resembling the morpho-functional units of mammalian dentitions (e.g., *Tupinambis teguixin*[36]). These differences among multicuspid dentitions allow for increased dietary specialisation, typically towards higher plant consumption.

We confirm here across the whole of Squamata the link noted previously between plant-eating squamates and a specialised, typically more complex dentition[17], similar to those hypothesised or discovered for early tetrapods[4], crocodyliforms[6], and mammals[5]. The generality of these findings suggests similar ecological and dietary selective pressures for complex dental phenotypes operate across all tetrapods. We find strong support for correlated adaptive evolution of multicuspidness and plant consumption in Squamata as a whole, and that both traits promoted increased diversification, with increased speciation coinciding with increasing tooth complexity and/or plant consumption in five major squamate groups (Egerniinae, Polyglyphanodontia, *Podarcis*, Pleurodonta, and a subclade of Iguanidae). Factors previously identified as influential for squamate diversification include global temperatures[37,38], latitude[39], and habitat[40,41]. In broad accordance with previous reports[42–44], we identify locally intense diversification dynamics and a diversity peak during the late Cretaceous, a phase shift at the Cretaceous–Paleogene (K–Pg) boundary, and an increase in diversity during the Cenozoic. We also note some discrepancies in more specific patterns such as the degree of Cenozoic diversification, likely stemming from differences in methods (tree-based vs. time series) and known biases like the pull of the Present. In agreement with other recent studies[45], we propose environmental factors such as the floral turnovers towards angiosperm dominance of the Cretaceous Terrestrial Revolution[46,47] and the subsequent evolution of angiosperm-dominated environments (notably rainforests) in the Cenozoic[48,49] as the most plausible drivers of increasing plant consumption in squamate evolution[50]. The Cretaceous saw the diversification of omnivores and herbivores among several tetrapod clades, including ornithischian and theropod dinosaurs[7–9], crocodyliforms[6], testudines[51], and multituberculates[13]. Specifically, during the KTR, squamate speciation locally peaked in the Campanian, and both extinction and turnover (here defined as the ratio of extinction to speciation rate) were overall highest. All three metrics drop in the Maastrichtian and across the Cretaceous–Paleogene boundary, a pattern reminiscent of the Maastrichtian decline in speciation rates recently identified in non-avian dinosaurs[52] and suggesting the K–Pg extinction event had less of an effect on squamate diversification dynamics than the KTR. These KTR diversification shifts coincide with the majority of the only period where reductions in both tooth complexity and plant consumption outnumber increases, suggesting a rapidly shifting set of available dietary niches as previously proposed for mammals of the same time[53]. Through the KTR, squamates experienced steadily increasing phenotypic disparity towards a peak in the latest Cretaceous[54], a period during which they were also arguably most ecologically innovative (see, e.g., mosasaurs[55,56], polyglyphanodonts[32]). Nevertheless, important ecological diversification also occurred during the Cenozoic,

which saw the most originations of plant consumption, though without resulting in a diversity of herbivores comparable to mammals. Reductions of squamate cusp number most often followed plant consumption reductions, suggesting relaxed selective pressures on diet enabled the loss of tooth complexity, as with mammals[29,30]. However, selective pressures on squamate teeth may not be as intense as for mammals. Most plant-eating squamates still consume insects[57], suggesting that, unlike in Mammalia, no hyper-specialist ratchet operated[58,59]. Besides, predatory squamates with single-cusped teeth—snakes in particular—also noticeably contributed to the diversification of the group during the Cenozoic[60], though without reaching the intensity of mammalian radiations[43,44]. This supports the contribution of factors other than tooth complexity and plant consumption to the diversification of certain squamate groups and emphasises the unique macroevolutionary dynamics of snakes, although key dental-dietary adaptations like the venom-delivering fangs of Colubroides[60–63], the ziphodont teeth of monitors[64–66], or the piercing, cutting, and crushing teeth of mosasaurs[56,67–69] may also have contributed to the success of these groups.

The patterns of squamate dental complexity evolution we observe offer a valuable counterpoint to the mammalian picture, exemplifying dental-dietary adaptations that responded to similar selective pressures, while resulting in more labile dental complexity throughout evolution. Despite vertebrates sharing a basic tooth gene-network[15], mammal teeth are more integrated structures, less prone, through developmental fine-tuning and intense selective pressures, to loss of complexity, though also capable of accumulating significantly more variance and reaching farther phenotypic extremes over time[70]. Since such finely tuned dental morphologies and precise occlusion have a critical role in ensuring mammals meet their high-energy needs[2], endothermy may limit the possibilities of mammalian dental simplification compared to ectothermic squamates. Several dental developmental differences to mammals can be suggested to explain why squamates did not fall into a developmental complexity trap[71], but instead evolved complex teeth highly liable to developmental instability and simplification[72,73]. These include simpler, less compartmentalised expression of dental development genes during tooth formation[16,19], a less complex morphological starting point than mammal teeth[12], and potentially simpler and/or looser gene regulatory networks[15]. We propose these characteristics of squamates explain both the evolutionary lability of their dental complexity and diet, and the near-complete absence of mammal-like teeth in over 250 million years of squamate history[32].

## Methods

**Phylogenies**. We gathered our own observations and reports from the literature on cusp number and diet for 548 species (429 extant species and 119 fossil species equally distributed between Mesozoic and Cenozoic). The data include all major squamate groups plus squamate stem taxa ($n = 545$)—including the oldest known squamate *Megachirella wachtleri*, two rhynchocephalians (the extant *Sphenodon punctatus* and the fossil *Gephyrosaurus bridensis*), and a stem lepidosaurian (*Sophineta cracoviensis*). To provide a phylogenetic framework for our analyses, we assembled an informal super-tree[74] for the 548 taxa. For topology we followed the total evidence phylogeny of Simões et al.[23]—the first work to find agreement between morphological and molecular evidence regarding early squamate evolution. The same source provided time calibrations for *Sophineta cracoviensis*, fossil and extant Rhynchocephalia, stem squamates and crown squamate groups. Using additional sources, we gathered complementary information on the stem and crown of Gekkota[75–77], Dibamia[77], Scincoidea[76,77], Lacertoidea[76–82], including Polyglyphanodontia[76,83], Mosasauria[76], Serpentes[76,77], Anguimorpha[76,77], and Iguania[76,77,84–86]. We followed Simões et al.[23] for the relationships of *Najash* and *Pachyrhachis*, and Pyron[76] for the placement of *Haasiophis* and *Eupodophis* (which were not sampled by Simões et al.[23]), thus reflecting both placements for simoliophiid snakes. To avoid over-sampling Liolaemidae, we randomly selected species according to relative abundance of dietary categories within the group[50] and of liolaemids among squamates. In the absence of time-calibrated phylogenetic information, we used temporal ranges from the Paleobiology Database (https://

www.paleobiodb.org) and checked accuracy by comparison with cited sources. Each squamate group stated above was grafted onto the backbone of the Simões et al.[23] phylogeny according to its proposed calibrations. Node calibrations falling within the 95% highest posterior density for the corresponding node in the Simões et al.[23] phylogeny were kept unchanged. Where a calibration fell beyond that range, the calibration of Simões et al.[23] was preferred. For taxa and nodes not included in Simões et al.[23] and with phylogenetic data lacking time-calibration, we used the code of Mitchell et al.[87,88] to generate calibrations based on last appearance dates and estimated rates of speciation, extinction, and preservation. The method—derived from Bapst[89]—allows the stochastic estimation of node age based on the inferred probability of sampling a fossil and probability density of unobserved evolutionary history, though nodes are sampled downwards towards the root rather than upwards from it. We used preliminary BAMM 2.6[87] runs not including the taxa concerned to generate estimates of speciation and extinction rates and selected a preservation rate of 0.01 (see below). Our tree includes 27 unresolved nodes, denoting phylogenetic uncertainty. For methods requiring a fully dichotomous tree, we used the function multi2di in ape 5.3[90] for R 3.6.1[91] to generate a random dichotomous topology. Because of the sensitivity of BAMM to zero-length branches, we then used the method of Mitchell et al.[88] to generate non-zero branch lengths in randomly resolved polytomies with fossil taxa. We used the same randomly resolved and calibrated tree in all analyses requiring a dichotomous tree. Node and branch nomenclature for the polytomous and dichotomous tree are available in Supplementary Figs. 1–4. We referred to the August 12th, 2019 version of the Reptile Database (http://www.reptile-database.org) and the Paleobiology Database (https://www.paleobiodb.org) for taxonomic reference of extant and fossil species (respectively).

**Dietary data**. We followed Meiri[92] and Pineda-Munoz & Alroy[93] for dietary classification. Accordingly, when quantitative dietary data were available, we classified species based on the main feeding resource in adults (i.e., >50% of total diet in volume[93]). Species consuming >50% plant material were classified as herbivores. We followed Meiri[92] and Cooper & Vitt[94] in defining omnivorous diets as including between 10 and 50% of plants, to account for accidental plant consumption by some predators. Among predators, carnivores are defined as feeding mostly on vertebrates. Predators consuming primarily arthropods and molluscs are insectivores. We could find no published dietary hypothesis for 64 out of 119 fossil species, which we assigned to the most plausible of our diet categories based on tooth complexity and the diets of closely related taxa (see Supplementary Data 3).

**Geometric morphometrics**. Specimens of 75 species were selected to represent all major groups of squamates with multiple-cusped teeth and based on the quality of the material available. We extracted two-dimensional outlines for geometric morphometric analyses (Supplementary Data 4) from 52 X-ray computed microtomography scans (microCT-scans), 12 photographs, 10 anatomical drawings of specimens, and one scanning electron microscopy (SEM) image. Sources included the literature, the Digital Morphology (DigiMorph) library, four new photographs, and four new microCT-scans (see below and Supplementary Data 3).

To analyse morphological variation of tooth shapes, we collected two-dimensional open outlines of a left upper posterior maxillary multicuspid tooth in labial view with ImageJ 1.47v[95]. We chose whenever possible the tooth with the most numerous cusps in the quadrant. If no left maxillary tooth was sampled or suitable for tracing an outline, we referred to the right quadrant or lower jaws and mirrored the outline adequately to retain the same orientation. We used the EqualSpace function from PollyMorphometrics 10.1[96] for Mathematica 10[97] to normalise teeth outlines as sets of 200 equally spaced points based on Bézier splines functions.

We used Momocs 1.3.0[98] for R[91] to perform geometric morphometric analyses of tooth outlines. We first applied a Bookstein alignment[99] and, for each outline, computed by Discrete Cosine Transform (DCT) the first 21 harmonic amplitudes[100]. Harmonic coefficients were then processed by Principal Component Analysis (PCA)[101]. We limited graphical representation of the PCA to its first two axes, accounting for 89% of all morphological variation. To determine the significance of our dietary grouping, we fitted a phylogenetic multivariate linear model using penalised likelihood (PL)[102,103] on all PC scores using mvMORPH 1.1.4[104] for R[91]. As we sampled only two carnivorous species, we added these to our insectivorous sample ($n = 50$), so making a predatory group, to avoid spurious conclusions arising from groups with extremely low-sample sizes. Model fit was performed using Pagel's $\lambda$[105] to jointly estimate the phylogenetic signal in model residuals. We then used a one-way phylogenetic PL-MANOVA to evaluate overall differences between dietary groups. To examine how dietary groupings best separate, we computed a discriminant analysis (DFA) on the regularised variance-covariance matrix of the fitted model. To test between-group differences, we used general linear hypothesis testing through contrast coding. We fitted a model for which each group was explicitly estimated to test compound contrasts. Lastly, we used mvMORPH to fit Brownian Motion (BM), early burst (EB), and Ornstein-Uhlenbeck (OU)[106] multivariate models of continuous trait evolution on the first five PC axes, representing over 95% of total variance and compared relative fit using Akaike weights[107] (see Supplementary Tables 10 and 11).

**Ancestral character state reconstructions**. We reconstructed the evolution of cusp number and diet using maximum likelihood (ML) ancestral character state reconstruction under a time-reversible continuous Markov transition model[108,109] as implemented in phytools 0.6–99[110]. We retrieved marginal ancestral states at the nodes of the tree with the re-rooting algorithm from the same package[111] and generated a model of character evolution by averaging three character transition matrices (all transitions allowed with either all rates different, symmetrical rates, or equal rates) according to their respective fit through Akaike-weights model averaging[107] (see Supplementary Tables 3 and 4). Finally, we used stochastic character mapping (1,000 simulations) to compute the most likely character states at each node based on the model-averaged transition matrix[112]. In contrast with tooth complexity ancestral states reconstructions, extant data allow the formulation of informed hypotheses on possible dietary transitions in squamates. Insects are an important food resource for the juveniles of many squamate species, and several extant species of plant consumers show an ontogenetic dietary shift from insectivorous juveniles to omnivorous or herbivorous adults[57,94,113,114]. Moreover, extant data show that predatory squamates may rely on plant material depending on environmental conditions[50,94,115–117]. Therefore, it has been hypothesised that squamate plant consumption originated in predatory animals, which evolved increasingly more plant-based diet through time under selective pressure[50,94]. We thus chose to test a specific hypothesis of dietary transitions against naive models and base our reconstructions on the best-performing model. We compared the respective fit of three default models (all transitions allowed) to three variants of our hypothesis of dietary transitions (limiting transitions to carnivore-insectivore, insectivore-omnivore, and omnivore-herbivore, with three transition rate regimes) and selected the model with highest relative fit (i.e., the custom model with all rates different) to retrieve ancestral states at the nodes (see Supplementary Tables 3 and 4). Through the same approach, we generated ancestral character reconstructions with three states (predatory, omnivorous, and herbivorous; see Supplementary Tables 8 and 9), which we then pruned to the geometric morphometric dataset to generate a mapping of regimes for a multi-peak OU multivariate model based on diet (see above; Supplementary Fig. 9). Based on the four-state diet and tooth complexity ancestral reconstructions, we gathered a list of changes in cusp number and plant consumption. Subsequently, we identified pairs of increases or decreases in both traits belonging to the same phylogenetic path (the unique succession of branches connecting a descendent lineage to one of its ancestors) and noted whether each initiated by a change in cusp number, plant consumption, or whether both changes happened on the same branch.

**Tests of correlated evolution**. We used BayesTraits 3.0.2 (www.evolution.rdg.ac.uk) to run Markov Chain Monte Carlo (MCMC) models of evolution of tooth complexity and diet with independent or correlated (i.e., assuming rates of transition in one trait depend on the character state of the other) rates of character transition. To improve rate estimations with our discrete dataset, in each run we scaled our tree to obtain an average branch length of 0.1 (i.e., scaling factor = 5.017e-3) as recommended by the software manual. Owing to method limitations, we transformed our discrete tooth complexity and diet characters into binary traits (teeth bearing one cusp vs. two cusps or more, and carnivores and insectivores (predators) vs. omnivores and herbivores (plant consumers), respectively). We computed two independent runs, each with four independent chains run for 110,000,000 iterations with default rate priors. We discarded the first 10,000,000 iterations as burn-in. We sampled parameters every 10,000 iterations and checked each chain for convergence (through visual examination) and large effective sample size (using CODA 0.19-3[118] for R[91]). We used a steppingstone sampler[119] to retrieve the marginal likelihood of each model (250 stones, each run for 10,000 iterations), which we compared with a log Bayes Factor to provide a measure of relative support of each model[120]. Analyses of the following combinations of different binarizations of the dataset yielded similar results, i.e., correlated evolution of cusp number and diet: one or two cusps vs. three cusps or more combined with carnivores and insectivores vs. omnivores and herbivores, and one to three cusps vs. more than three cusps combined with herbivores vs. other diets. As expected, a correlated model was weakly or not supported for other combinations (Supplementary Table 6).

**Rates of phenotypic evolution**. We estimated the evolutionary rate of tooth shape change through the variable rates model of BayesTraits 3.0.2[121,122]. In this approach, a reversible-jump Markov Chain Monte Carlo (rjMCMC) algorithm is used to detect shifts in rates of continuous trait evolution—modelled by a Brownian motion (BM) process—across the branches of a phylogenetic tree. This is achieved by estimating the location of the shifts in rates (the product of a homogeneous background rate with a set of rate scalars) by using two different proposal mechanisms (one updating one branch at a time and one updating complete subclades). We used the default gamma priors on rate scalar parameters. Support for rate heterogeneity was then further confirmed by comparing the fit of the variable rates model against a null single-rate Brownian model. Here, we ran a variable rates model and a homogeneous Brownian model on the scores of the first 12 pPC axes from our phylogenetic PCA of tooth outlines, accounting for over 99% of the total variance. As PC axes can be correlated in a phylogenetic context, we

used the phylogenetic PC scores to remove evolutionary correlations[123,124]. We ran a phylogenetic principal component analysis (pPCA)[124] on the first 21 harmonics obtained by DCT using phytools 0.6–99[110] for R[91]. As for the original PCA, we found the two first pPC axes accounted for the largest part of all morphological variation (83% of cumulative variance). All parameters used were the same as for the correlated evolution tests (see above): two independent runs, four independent chains per run, 110,000,000 iterations, 10% burn in, default priors, rescaling factor = 3.206e-3, sampling every 10,000 iterations, convergence and sample size checks, steppingstone sampler with 250 stones run for 10,000 iterations. Finally, we plotted our species tree (using phytools 0.6–99[110], ggtree 1.8.1[125], and viridis 0.5.1[126] for R[91]) with branches scaled by the averaged rate scalars across posterior samples (returned by the Variable Rates Post Processor; http://www.evolution.reading.ac.uk/VarRatesWebPP/), thus indicating the relative deviation from the background rate of change.

Likewise, we used a variable rates model approach on discrete data to detect heterogeneity in character transition rates for tooth complexity and diet. The variable rates model operates on discrete data by breaking the assumption of a single character transition rate matrix defined for the entire tree, which it achieves by rescaling this transition matrix in different parts of the tree using an rjMCMC algorithm. As for continuous data, the process generates a posterior distribution of scalars for each branch, and comparison with a null MCMC model with a constant transition matrix allows evaluation of support for heterogeneity in the strength of character transition rates. We ran the variable rates and null models similarly to tooth shape data (see above), using binarized tooth complexity and dietary data to avoid over-parameterisation. The variable rates post processor returned the averaged branch rate scalars used to plot the tree according to local deviations from the background transition matrix. The large variances returned by the post processor for some rate scalars, however, denote a relatively complex model to fit and warrant adequate caution in interpreting absolute rate scalar values, though relative rate differences should be fully representational. For clarity, we coloured each branch according to a common log-transformed scale. We again tested alternative binarizations of diet and tooth complexity and found support for a variable rates model for the alternative binarization of diet and one other binarization of tooth complexity (one or two cusps vs. three cusps or more) (Supplementary Table 7).

**Models of diversification**. We fitted different trait-dependent models of speciation and extinction (BiSSE, HiSSE) and associated trait-independent null models with hisse 1.9.5[127] for R[91], for which we compared relative fit using Akaike weights[107]. In addition, we used an rjMCMC algorithm with BAMM 2.6[87] and BAMMtools 2.1.6 for R[91] to model rates of speciation and extinction independently from trait evolution on a random resolution of our super-tree (see above). This is currently the only available method allowing branch-specific estimation of diversification rates on non-ultrametric trees (i.e., including fossil taxa) by using a fossilised birth-death process[87], whereas more recent and more robust approaches such as CladS[128] cannot handle them. The choice of a tree-based method over approaches based on time series (such as PyRate[129]) allows the direct comparison of diversification rates with Maximum Likelihood ancestral state reconstructions and branch-specific rates of phenotypic evolution (see above) and provides an assessment of diversification dynamics not only through time, but across clades. This removes ambiguity in defining time series for different clades while simultaneously coping with issues of poor fossil records. We produced ten independent replicates, each with four independent chains run for 20,000,000 generations. We used priors generated by the setBAMMpriors function of BAMMtools (expectedNumberOfShifts = 1.0; lambdaInitPrior = 10.2511119939331; lambdaShiftPrior = 0.00400165322948176; muInitPrior = 10.2511119939331), a preservation rate prior of 0.01 to reflect the sampling biases affecting the squamate fossil record[42] in the absence of any published or reported estimation, and a global sampling fraction of 0.048 accounting for our sampling relative to the total diversity of living and extinct squamates referenced in both the Reptile Database (http://www.reptile-database.org) and the Paleobiology Database (https://www.paleobiodb.org). We set a 10% burn-in and checked convergence through visual examination and effective sample size with CODA 0.19-3[118]. As we encountered many equiprobable configurations, for each run we computed the maximum shift credibility (MSC) configuration and extracted speciation and extinction rates for clades defined by each node immediately above a shift, plus mean rates outside these clades (background rate). We then calculated a mean shift magnitude for each clade using the ratio of its mean speciation rate over the mean background rate[130]. To control for the influence of aquatic taxa during the KTR, we repeated analyses on a tree devoid of Cretaceous aquatic taxa (ten mosasaurs and three snakes) and found no changes to our results. To allow the comparison of shifts in diversification rate (inferred along branches) and character transitions (reconstructed at nodes only), we assigned each shift to the node immediately above it (Supplementary Tables 12 and 13).

**Statistics**. We performed all univariate non-parametric tests using rcompanion 2.3.25 (https://www.rcompanion.org) and the base stat package in R 3.6.1[91]. All effect sizes[131] and their 95% confidence intervals were computed by bootstrap

over 10,000 iterations. Sample size for all tests is $n = 548$. A Kruskal–Wallis $H$ test[132] on tooth complexity levels among squamate dietary categories showed a statistically significant effect of diet on the level of tooth complexity ($\chi^2 = 144.27$, df = 3, $p$-value = 4.5e-31, $\varepsilon^2 = 0.26$ [0.20, 0.34]). Post hoc pairwise two-sided Wilcoxon–Mann–Whitney tests[133,134] showed statistically significant differences between all dietary categories (see Supplementary Table 1 for full reporting).

We used mvMORPH 1.1.4[104] for R[91] to perform regularised phylogenetic one-way multivariate analyses of variance (MANOVA) and multivariate general linear hypothesis tests in a penalised likelihood framework[102,103]. For each test, we assessed significance with a one-sided Pillai trace test over 10,000 permutations of the Pillai trace[135] obtained through regularised estimates[102,103]. A regularised phylogenetic one-way MANOVA on the principal component scores of 75 tooth outlines showed statistically significant differences in two-dimensional (2D) tooth shape between diets ($V = 1.04$, $p$-value = 0.001). We then used general linear hypothesis tests to evaluate simple and compound contrasts between groups, of which all but one were statistically significantly different (see Supplementary Table 2 for full reporting).

Two-sided Wilcoxon–Mann–Whitney tests[133,134] on speciation and extinction rates inferred using the best-performing trait-dependent model of diversification (see Supplementary Table 14 and Supplementary Fig. 12) show multiple-cusped taxa have both statistically significantly higher speciation and extinction rates than taxa with single-cusped teeth. Likewise, plant-consuming (i.e., omnivorous and herbivorous) taxa have both statistically significantly higher speciation and extinction rates than mainly predatory taxa (i.e., carnivores and insectivores) (see Supplementary Fig. 12 for full reporting).

Unless specified otherwise, all analyses and tests were replicated twice independently (ten independent replicates for the trait-independent diversification model) with consistent results.

**Photographs and X-ray computed microtomography**. Photographs of ten specimens were captured at the Museum für Naturkunde (Berlin, Germany). New microCT-scan data was generated for 24 specimens using a Skyscan 1272 microCT (Bruker) at the University of Helsinki (Finland), a Skyscan 1172 microCT (Bruker) at the University of Eastern Finland (Kuopio, Finland), and a Phoenix nanotom CT (GE) at the Museum für Naturkunde (Berlin, Germany). Three-dimensional surface renderings were generated using Amira 5.5.0[136].

**Specimen collection**. Specialised retailers provided specimens of five species (see Supplementary Data 3). The Laboratory Animal Center (LAC) of the University of Helsinki and/or the National Animal Experiment Board (ELLA) in Finland approved all reptile captive breeding (license numbers ESLH-2007-07445/ym-23 and ESAVI/7484/04.10.07/2016).

**Art credits**. Figure 1a (top to bottom): Steven Traver (CC0 1.0), F.L. after Ted M. Townsend (silhouette drawn from photograph, CC-BY 2.5 https://creativecommons.org/licenses/by/2.5/deed.en), David Orr (CC0 1.0), F.L. after Darren Naish (used with permission), Alex Slavenko (CC0 1.0), Iain Reid (used without modification, CC-BY 3.0 https://creativecommons.org/licenses/by/3.0/), T. Michael Keesey (CC0 1.0), Steven Traver (CC0 1.0), T. Michael Keesey (from a photograph by Frank Glaw, Jörn Köhler, Ted M. Townsend & Miguel Vences, used without modification, CC-BY 3.0 https://creativecommons.org/licenses/by/3.0/). Figure 2 (counterclockwise from 90°): F.L. after Darren Naish (used with permission), F.L. after Dick Culbert (silhouette drawn from photograph, CC-BY 2.0 https://creativecommons.org/licenses/by/2.0/deed.en), Alex Slavenko (CC0 1.0), T. Michael Keesey (from a photograph by Frank Glaw, Jörn Köhler, Ted M. Townsend & Miguel Vences, used without modification, CC-BY 3.0 https://creativecommons.org/licenses/by/3.0/), Michael Croggie (CC0 1.0), Jack Meyer Wood (CC0 1.0), F.L. after Scott Robert Ladd (silhouette drawn from photograph, CC-BY 3.0 https://creativecommons.org/licenses/by/3.0/). See http://phylopic.org/about/ and https://creativecommons.org/licenses/ for additional license information.

**Reporting summary**. Further information on research design is available in the Nature Research Reporting Summary linked to this article.

## Data availability
All datasets generated and analysed during the current study (tip-state dataset, polytomous and dichotomous versions of our phylogeny, 2D outlines) are available as Supplementary Data files 1–4. We used the Reptile Database (http://www.reptile-database.org) to access taxonomic information on extant species and the Paleobiology Database (https://www.paleobiodb.org) for taxonomy and temporal ranges of fossil species. Dietary data were extracted from the database of Meiri[92] and the published literature (see Supplementary Data 3). CT-scan data are in part publicly available on the Digimorph database (http://digimorph.org/) and the published literature (see Supplementary Data 3). The remnant CT-scan data are available through N.D.-P., upon reasonable request. Source data are provided with this paper.

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

## Acknowledgements

We thank Ilpo Hanski and Martti Hildén (Luonnontieteellinen keskusmuseo, Helsinki, Finland) for specimen loans, Johannes Müller (Museum für Naturkunde, Berlin, Germany) for specimen loans and access to collections and CT-scanning facilities, Jessie Maisano (University of Texas, Austin, TX) for sharing data from the DigiMorph database, Arto Koistinen (University of Kuopio, Finland) and Heikki Suhonen (University of Helsinki, Finland) for access to CT-scanning facilities, Arto Koistinen, Simone Macrì, Kristin Mahlow, and Filipe Oliveira da Silva for acquiring morphological data, as well as Jukka Jernvall, Mikael Fortelius, and the Helsinki Evo-Devo community for discussions. We thank Vincent Bonhomme, David Caetano, Andrew Meade, and Johnathan Mitchell for their help in implementing Momocs, HiSSE models, BayesTraits, and BAMM 2.6, respectively. We also thank Robert Espinoza for precisions on liolaemid diets. This work was supported by funds from the Integrative Life Science doctoral programme (ILS; to F.L.), the Center for International Mobility scholarship programme (CIMO; to F.L.), the University of Helsinki (to N.D.-P.), the Institute of Biotechnology (to N.D.-P.), Biocentrum Helsinki (to N.D.-P.), and the Academy of Finland (to N.D.-P.).

## Author contributions

F.L., I.J.C. and N.D.-P. designed the experimental approach. F.L. and N.D.-P. collected the specimens for microCT-scanning. F.L. character-coded species from the literature and specimen data. F.L. collected tooth outline semi-landmark data. F.L. performed the research. F.L. analysed the data, with contribution from J.C., I.J.C. and N.D.-P. F.L. made the figures. F.L. produced the first draft, and F.L., J.C. and I.J.C. wrote the paper, to which all authors contributed in the form of discussion and critical comments. All authors approved the final version of the manuscript.

## Competing interests

The authors declare no competing interests.
