## [Peer Review File · Nature Communications]

Multiple evolutionary origins and losses of tooth complexity in squamatesREVIEWER COMMENTS

Reviewer #1 (Remarks to the Author):

Remarks to authors

GENERAL COMMENTS

The manuscript provides a detailed assessment of the rates of dental shape evolution in squamates based on geometric morphometric data and tests specific points in the squamate tree of life where tooth complexity increased and decreased, along with reconstructed changes in dietary habits. This is followed by estimates of speciation and extinction rates in squamate evolution. The authors propose that squamate tooth complexity evolved multiple times independently, with several cases of reversals towards reduction in complexity, and later reacquisition of lost complexity. This is a more labile system than in mammals and, according to the authors, changes in diet (towards herbivory) and tooth complexity were one of the key drivers of squamate diversification.

This manuscript addresses important questions in squamate evolution and the authors put considerable effort in data collection, analysis of character and species evolution, and trying to characterizing tooth shape evolution. The results can be of considerable impact to understand squamate evolution. However, before recommending this manuscript for publication, there are important issues that need to be addressed (for further details, see detailed comments below):

- 1) Formatting: this manuscript seems to have been transferred from Nature, and so its formatting is at odds with how it should be formatted for Nature Communications. The main text is currently extremely short and, given the much extra space provided by NC, I would strongly recommend expanding it to further develop on their introduction, results, discussions and conclusions. Importantly, there are key areas of the methods and results that need to be expanded by providing further important details of their results.
- 2) The authors should make it explicit the rationale for their chosen method to estimate diversification rates in squamate evolution.
- 3) The authors result of diversification rates contrast with some recently published estimates of squamate species richness through time during the Mesozoic and Cenozoic. They should address those differences and suggest potential reasons for such discrepancies.
- 4) There seems to be a mismatch in their figure results between major changes in tooth complexity and major shifts in speciation rates, contrary to their major conclusion.

DETAILED COMMENTS

MAIN TEXT

Lines 89-93: please name the 10 major clades with independent acquisition of plant consumption (either listed in the text or as a table), as you did with the six major clades of increased complexity.

Lines 121-129: the authors mention the identification of eight points of speciation increase coinciding exactly/one note above points of increase in tooth complexity or plant consumption. However, comparing figs. 2,4a,b, (points of increasing tooth complexity) with fig. 4c (points of speciation rate shifts), those do not coincide. High speciation is depicted in early squamates, scincids, lacertids, derived snakes, Polyglyphanodontidae, Mosasauria, varanids, and stem iguanians, none of which coincide with points of increase in tooth complexity. Even the points of detected increased speciation within iguanians occur further crownward in the tree when compared to the point of increased those complexed in iguanian evolution (with the exception of agamids).

As a general challenge to the authors' conclusions, and that they should address, is the well-known fact that some of those lineages with high diversity, such as varanids and derived snakes (i.e. colubroids), have some of the simplest dental shapes among squamates. The Varanus and colubroid radiations represent the most successful taxonomic radiations in squamate evolution, along with the Anolis radiation within iguanians—e.g. (Pianka & Vitt, 2003), clearly indicating that changes in diet and tooth complexity do not relate to their ecological and evolutionary success.

Lines 131-135: what are those unobserved factors that may have contributed to diversification? You mentioned here "(Extended Data Table 4 and Figure 5)", but those do not make explicit mention of additional factors. Also I believe he referred to Supplementary data figure 5, as there is no Figure 5.

Lines 157-159: only two clades are listed among the ones where the authors find a strong support for correlated evolution of dental shape and herbivory (also associated with high speciation). As mentioned elsewhere here, please list all of those clades somewhere in the main text. This is one of the key findings of this manuscript and is key to illustrate their findings and for all readers interested in squamate evolution.

Lines 157-165: the authors find increasing levels of speciation rates (and net diversification) in squamates between the Early Jurassic and the end of the Cretaceous, with only one visible drop in those levels at around 125Ma during the Early Cretaceous. Those results are interesting, but come into strong conflict with the measures of species richness through time using data from the fossil record published over the last two years that suggest low levels of squamate diversity during most of the Mesozoic and a considerable increase in the squamate diversity only at the Late Cretaceous (Cleary et al., 2018; Close et al., 2020; Close et al., 2019). The authors results here are derived from fundamentally different methodologies, but given the fundamental differences between their results in previous ones (with a considerable impact to our understanding of squamate evolution in deep time) I strongly suggest the authors to address such differences in their main text, and potential reasons for those differences.

My impression is that the poor fossil record of squamates from the early Triassic to the late Cretaceous (owing to low research effort, geographical biases in fossil collection, etc) creates biases in diversification dynamics estimated using fossil data with the methods employed in the studies cited above (e.g., subsample diversity data quorum). Perhaps alternative methods with stronger corrections for fossil biases (BAMM and PyRate), such as employed by the authors, are more capable to avoid biases introduced by the poor early fossil record of squamates. However, see also my comments below on the methodology section related to BAMM and phylogeny-based estimates of diversification.

Line 163/figure 4d: "net turnover". I believe you referring here to net speciation (speciation – extinction). "Turnover", or relative extinction, is something different.

FIGURES and TABLES

Figure 4c: please list the clade names for the clades with considerable speciation rate shifts (A-M) in the figure legend. Alternatively, least those 13 clade names on the side of figure 4c itself. Since nature communications allow up to 10 figures, you can easily split figure 4 so that component 4c is its own figure with enough space available to list those clades.

I noticed those clades are numbered in Extended Data Table 4, but I would also suggest replacing those numbers for clade names in the table, as the numbers are hard to track without a translation table for clade names.

Figure 4d. This is a key figure where the authors present the results of their speciation and extinction rates through time. Yet, this is a very small component of what is already a figure component. I suggest adding at least a much larger version of this figure as a supplementary figure so readers can see this figure in much greater detail, or a new figure of its own in the main text.

In the supplementary information, tables are provided to indicate the branches of the tree where dietary and tooth complexity transitions are happening, but the authors do not provide a figure with the nodes plotted in the tree so the readers can track down those node numbers in the tree. I strongly recommend adding that to the supplementary information, even if you need to break up the tree across three or four separate figures (and pages) because of the size of the tree. This makes it much easier for readers to directly assess the particular clades where those transitions are happening, without having to download the nexus tree files, import them into R, etc.

METHODOLOGY

Phylogenetic supertree (full species tree provided in the supplementary information): Different squamate systematists will definitely have their own preferences over specific aspects of the tree, an issue that is always impossible to avoid. However, I find that the overall topology and calibration times for their tree looks quite reasonable and I believe most systematists would agree with the overall tree structure.

However, there is one aspect of the tree that I find it necessary to point out as it will be highly controversial for anyone familiar with snake evolution: the paraphyly of simoliophiids. Every squamate phylogeny including fossil snakes that I am aware of including two or more species of simoliophiids always recovers simoliophiids as monophyletic. Currently, the authors place Haasiophis+Pachyophis in a distinct location from Pachyrhachis, further up in the tree. Although the placement of simoliophiids in snake phylogenies is constantly shifting, one thing everybody agrees on is on their monophyly (wherever they may be in the phylogeny of snakes). Having said that, the fact that all snakes have pretty much the same dental morphology will not affect their results on the rates of morphological evolution. But since this is a very controversial reconstruction of early snake phylogeny, I strongly recommend the authors to, at least, address this issue in their methods.

Geometric morphometrics: the PCA analysis and morphospace plotting (Fig. 1c) is important to demonstrate the overall distribution of variance in the data, but an actual discriminant analysis between groups (CVA or between group PCA) would be a better approach to demonstrate significant separation between the groups.

Tests of correlated evolution (lines 450-455): for the analysis with BayesTraits, in a similar manner to the BAMM analysis below, authors should explicitly specify parameters for their Bayesian analysis. The authors should indicate the number of independent runs, number of chains in each run, and what are the diagnostic reports for convergence and stationarity.

Models of Diversification (lines 505-525): the authors utilized BAMM v. 2.6 for their inferences on the shifts of speciation and extinction rates. As the authors correctly indicate, this is the most complete method to infer such rates from phylogenetic trees, as it includes the assumption of fossil sampling and fossilization rates. However, the authors do not indicate why they chose a phylogeny-based approach to estimate rates of speciation and extinction to begin with. Competing approaches, especially the latest version of PyRate (Silvestro et al., 2019), also provide estimates of speciation and extinction that use the vast amount of fossil record data available from the Paleobiology Database, and which are independent of potential variations in phylogenetic tree shape. It is not yet clear which of the two approaches is more accurate, but it would be definitely important for the authors to lay out their rationale for choosing BAMM to estimate speciation and extinction rates as opposed to other approaches, especially PyRate.

One specific point I believe authors should address is whether they believe that their rates of extinction may be affected by using a phylogeny-based approach. This is a common problem for extant phylogenies based on molecular data (Quental & Marshall, 2010; Silvestro et al., 2018), which in theory, the authors may have circumvent by using fossil data and BAMM v 2.6. Yet, the results presented here seem to have exceptionally low levels of extinction between 250-150Ma. If this extinction level is being underestimated, this could explain the steady increase in net diversification in squamates during the Jurassic, which contrasts to the results presented by recent studies using the fossil record (see also comments above for Lines 157-165).

For their implementation of BAMM, it is very important for the authors to provide some additional details on their implementation. For instance, they indicate they used priors generated by BAMMtools, but they do not specify what are, specifically, those prior choices. Additionally, they choose a preservation rate of 0.01 for the squamate fossil record, but do not indicate how they came up with this particular value. Please, elaborate more on those aspects as they are crucial to BAMM results.

Statistics (line 544): ..." Macroevolutionary rates...": Please replace that by speciation and extinction rates. Macroevolutionary rates, or rates of evolution, can be used for either diversification dynamics as well as rates of character evolution.

References cited above:

CLEARY, T. J., BENSON, R. B., EVANS, S. E. & BARRETT, P. M. (2018). Lepidosaurian diversity in the Mesozoic–Palaeogene: the potential roles of sampling biases and environmental drivers. *Royal Society open science* 5, 171830.

CLOSE, R. A., BENSON, R. B., ALROY, J., CARRANO, M. T., CLEARY, T. J., DUNNE, E. M., MANNION, P. D., UHEN, M. D. & BUTLER, R. J. (2020). The apparent exponential radiation of Phanerozoic land vertebrates is an artefact of spatial sampling biases. *Proceedings of the Royal Society B* 287, 20200372.

CLOSE, R. A., BENSON, R. B. J., ALROY, J., BEHRENSMEYER, A. K., BENITO, J., CARRANO, M. T., CLEARY, T. J., DUNNE, E. M., MANNION, P. D., UHEN, M. D. & BUTLER, R. J. (2019). Diversity dynamics of Phanerozoic terrestrial tetrapods at the local-community scale. *Nature Ecology & Evolution* 3, 590-597.

PIANKA, E. R. & VITT, L. J. (2003). *Lizards: Windows to the Evolution of Diversity*. University of California Press.

QUENTAL, T. B. & MARSHALL, C. R. (2010). Diversity dynamics: molecular phylogenies need the fossil record. *Trends in ecology & evolution* 25, 434-441.

SILVESTRO, D., SALAMIN, N., ANTONELLI, A. & MEYER, X. (2019). Improved estimation of macroevolutionary rates from fossil data using a Bayesian framework. *Paleobiology* 45, 546-570.

SILVESTRO, D., WARNOCK, R. C., GAVRYUSHKINA, A. & STADLER, T. (2018). Closing the gap between palaeontological and neontological speciation and extinction rate estimates. *Nature Communications* 9, 1-14.

Reviewer #2 (Remarks to the Author):

The authors present a study that describes the evolutionary history of dentition in reptiles. They present several lines of evidence showing that complexity in dentition is related to diet and has gone through transitions across the reptile lineage. Some of these transitions include evolving more complex or losing complexity in teeth morphology (namely, cusps). They postulate that changes in dentition and diet also contribute to the diversification of the reptile lineage.

In my opinion, the study is strong, analyses are appropriate, and their claims are validated via several lines of evidence. I'm incredibly impressed by the study and I failed to find any large issues that would prevent the study from being published. The PCMs used are what is standard in the field and their interpretations are reasonable, which is refreshing as many studies I have reviewed in the past have a hard time with PCMs and what they mean.

My one criticism of the study, and this is minor, is that I would like to see the authors explicitly test the hypothesis that dentition complexity is related to diet using an OU method (SLOUCH, for example) and present the adaptive landscape to provide further evidence that there is an adaptive optima for each kind of dentition and that the adaptive landscape is shaped by diet. This should be relatively quick to do and only add a few lines to the manuscript but would ease my concern that there are real peaks and valleys in the landscape and that dentition complexity is near an adaptive peak.

Otherwise, I commend the authors on the tremendous work they put into the study and believe this study will be of great value to the scientific community.

Reviewer #3 (Remarks to the Author):

Review, Multiple evolutionary origins and losses of tooth complexity in squamates

By Fabien Lafuma, Ian J. Corfe, Julien Clavel, Nicolas Di-Poï

Summary.

Lafuma et al. reconstruct patterns of tooth evolution in Squamata, one of the most diverse tetrapod clades, and compare these patterns to Mammalia. Insofar as tooth evolution in Mammalia is thought to be an important part of the success of the clade, studying a parallel radiation and comparing and contrasting the dynamics is intriguing, and they find distinct macroevolutionary dynamics (higher rates of complexity acquisition- and loss- but it seems less complexity than achieved by mammals). The paper is well written and seems solid and the conclusions are of broad general interest to people interested in macroevolution, rather than just lizard evolution, and therefore appropriate for publication for a high profile journal such as Nature Communications. In my opinion it's publishable with minor revisions although I'm more a morphology guy than a stats/tree guy, so others might have thoughts on the methods.

I think my one suggestion would be that it might be better to try to somehow map out these evolutionary events over time, perhaps by (1) plotting the changes in cusp number (increases and decreases in cusp count) over geological time, and/or (2) trying to look at tooth disparity (even just simple variables like crown width and cusp count) over time. I think doing this would help better communicate the results and the story. Some photos of representative tooth morphologies (as it stands, the emphasis on trees and plots makes it feel very abstract) might help provide context to the reader, as well as a bit more discussion of the kinds of morphologies seen in squamate teeth (which are more diverse than one might think). These are suggestions and my recommendation to accept isn't contingent on them.

NL

Detailed comments.

I think a little more detail on how squamates achieved multicuspid teeth might be useful, perhaps even figures or photographs showing the morphology. Just counting cusps doesn't really communicate the variety of tooth morphologies in lizards; there's more going on than one might think at first. For example, the polyglyphanodonts achieve tooth complexity several different ways. Polyglyphanodon expands the crown transversely to create transverse blades (this may or may not be related to the weird multicuspid arrangement seen in *Peneteius*). However, *Macrocephalosaurus* achieves tooth complexity by expanding teeth anteroposteriorly and evolving many small cusps, like a modern iguana. I think a nice summary figure- photographs showing the range of tooth morphologies seen in squamates- would also help communicate this? Some discussion might be useful as well.

Also, dolichosaurs, specifically *Coniasaurus*, have weird teeth and arguably you could call it bicuspid (the anterior carina is strongly convex in *Coniasaurus*) so that may be another evolution of multicuspid teeth?

"we reconstruct the MRCA of Squamata as unicuspid and infer at least 24 independent acquisitions of multicuspidness in squamate lineages." Sorta burying the lead here! This is really striking. Even if squamates never achieved the complexity seen in mammals, they evolve multicuspid teeth far more often, it seems, and in that sense they're very innovative. This is a really striking result and I think should be up in the abstract.

"while resulting in more labile dental complexity throughout evolution. "

Well it's like mammals didn't do it often, but did it really well, versus lizards which did it often but never quite to the extreme seen in mammals; even the most elaborate lizard teeth arguably aren't as complex as typical mammal dentition. So perhaps lizards were more evolutionarily versatile, but mammals achieved higher complexity. Maybe the two- complexity and versatility- are difficult to reconcile? We can argue about which is better. There are more lizard species, but mammals have higher biomass, probably occupy more niches (e.g. whales, bats, elephants have no squamate competitors).

Archosaurs seem sort of intermediate? The teeth of ceratopsians, hadrosaurids etc. are more complex than those of lizards arguably, less so than mammals in my opinion. They didn't evolve complex teeth all that often- there's a single clade, Ornithischia, that has most of the complex teeth of the entire group, with some tooth complexity in *Notosuchia*.

But arguably at some level how often you evolve complexity is less relevant than how well you do it? The eukaryotic cell and oxygenic photosynthesis are one-off innovations (like the tribosphenic molar) but they revolutionized life on Earth; they didn't evolve often but it made a huge difference. Humans (social animals with sophisticated tool use and language and giant brains) are a one-off event but have completely reshaped the biosphere.

“mammal teeth are more integrated structures, less prone, through intense selective pressures, to loss of complexity”

That’s an interesting problem. I guess the question is, are they intrinsically less evolvable due to developmental and anatomical constraints, or just exposed to different selective pressures? You do get homodont, unicuspid teeth in certain lineages- odontocete whales, armadillos. Numbats (*Myrmecobius*) do a weird thing where the teeth are functionally homodont/unicuspid like a lizard, but this is achieved by aligning the three cusps of the molars to form the functional equivalent of three, unicuspid teeth... which seems to suggest that mammals do struggle to evolve unicuspid teeth, even faced with the strong selective pressures for an insectivorous diet, they’re cheating by making a functionally unicuspid arrangement out of multicusped teeth. Just as often mammals seem to lose the teeth entirely (pangolins, anteaters, various cetaceans) which offhand I don’t think squamates ever did?

To me this suggests a tradeoff between complexity and evolvability. Mammals have achieved extremely high levels of tooth complexity, but perhaps the various developmental pathways needed to make a placental molar now limit the directions you can take it?

Re: the K-Pg boundary it’s a little surprising to see that more doesn’t happen there. In part, I think it’s extremely difficult to time some of these radiations and better models and calibrations may yet push some of these radiations into the Cenozoic (e.g. crown Pleurodonta) but even so, this wouldn’t affect things a lot. That is, even if crown Pleurodonta is a Paleogene radiation, they already had multicuspid teeth in the Cretaceous, and we don’t see a lot of acquisition in the basal branches of that tree.

This does sort of fit with my sense that lizards failed to innovate the way mammals do in the Cenozoic. Many mammal lineages shift from omnivory to herbivory, but lizards seem far more conservative in the Cenozoic- arguably they less innovative in the Cenozoic than in the Cretaceous where we had for example, mosasaurs, various polyglyphodont lineages evolving radically different tooth morphologies.

With respect to the discussion it might be useful to discuss squamates in terms of what else is happening in the Mesozoic, in other clades?. This is sort of touched on (the KTR) but note that we also have other groups experimenting with herbivory at this time. E.g. lots of herbivorous theropods- Oviraptorosauria, Ornithomimidae, Therizinosauroidea, even certain Troodontidae- show up in the Cretaceous. Highly specialized, herbivorous multituberculates also diversify, e.g. *Meniscoessus*. Herbivorous crocodylomorphs, like *Simosuchus* appear. Herbivorous turtles, e.g. *Basilemys*. It’s a big time for many small herbivores, not just lizards.

Minor issues: “do not” rather than “don’t” in the abstract (“don’t” and other contractions being more informal)

Multiple evolutionary origins and losses of tooth complexity in squamates

F. Lafuma, I. J. Corfe, J. Clavel, N. Di-Poi

Point-by-point replies to the reviewers' and editorial comments and suggestions

We thank the three reviewers for the positive evaluation of our manuscript. We have carefully considered all the comments and recommendations and explain below how we revised the entire manuscript to comply with these observations. We reproduce the referees' comments in italics and our responses and explanations are in plain text.

Replies to Reviewer #1:

GENERAL COMMENTS

The manuscript provides a detailed assessment of the rates of dental shape evolution in squamates based on geometric morphometric data and tests specific points in the squamate tree of life where tooth complexity increased and decreased, along with reconstructed changes in dietary habits. This is followed by estimates of speciation and extinction rates in squamate evolution. The authors propose that squamate tooth complexity evolved multiple times independently, with several cases of reversals towards reduction in complexity, and later reacquisition of lost complexity. This is a more labile system than in mammals and, according to the authors, changes in diet (towards herbivory) and tooth complexity were one of the key drivers of squamate diversification.

This manuscript addresses important questions in squamate evolution and the authors put considerable effort in data collection, analysis of character and species evolution, and trying to characterizing tooth shape evolution. The results can be of considerable impact to understand squamate evolution.

We thank the reviewer for their thorough comments, which have greatly helped us to improve the significance and readability of the manuscript. We agreed with most comments and suggestions and have revised the text, Figures and Supplementary Information accordingly.

However, before recommending this manuscript for publication, there are important issues that need to be addressed (for further details, see detailed comments below):

1) Formatting: this manuscript seems to have been transferred from Nature, and so its formatting is at odds with how it should be formatted for Nature Communications. The main text is currently extremely short and, given the much extra space provided by NC, I would strongly recommend expanding it to further develop on their introduction, results, discussions and conclusions. Importantly, there are key areas of the methods and results that need to be expanded by providing further important details of their results.

We agree that the longer formatting allowed by *Nature Communications* warrants an extension of the manuscript. The revised version is more detailed in all parts, notably regarding key areas identified by all three reviewers (see below).

2) The authors should make it explicit the rationale for their chosen method to estimate diversification rates in squamate evolution.

Done. We have made more explicit our choice of BAMM for estimating diversification rates from non-ultrametric trees – *i.e.*, trees including both extant and extinct species (see below and revised Methods, manuscript lines 543-552).

3) The authors result of diversification rates contrast with some recently published estimates of squamate species richness through time during the Mesozoic and Cenozoic. They should address those differences and suggest potential reasons for such discrepancies.

Our tree-based diversification modelling method is inherently distinct from the time-series approaches cited in our manuscript (Cleary *et al.* 2018; Close *et al.* 2019, 2020) and by the reviewer (Silvestro *et al.* 2019), which can explain the different diversification dynamic we recover (notably an exponential increase in speciation through the Cenozoic). We address these differences and possible biases like the pull of the Present in our reply below as well as within the revised manuscript's discussion (lines 227-232).

4) There seems to be a mismatch in their figure results between major changes in tooth complexity and major shifts in speciation rates, contrary to their major conclusion.

After re-examination, it appeared a column of Extended Data Table 3 detailing these results was accidentally cropped out during formatting. The full table is now shown in the revised manuscript (see Supplementary Table 12). We may also have failed to unambiguously present

these results; we further detailed our reasoning in our reply below. Nevertheless, we reiterate our interpretation of the result and maintain our claim that several shifts in speciation directly coincide or are found within one node of an increase in tooth complexity and/or plant consumption.

MAIN TEXT

- *Lines 89-93: please name the 10 major clades with independent acquisition of plant consumption (either listed in the text or as a table), as you did with the six major clades of increased complexity.*

Done. As proposed by the reviewer below (see comments on Fig. 4c), the names and node numbers of all major clades evolving tooth complexity and plant consumption are now indicated in the revised Fig. 2 and its caption and panel b of Supplementary Fig. 6 (Extended Data Fig. 3 of the initial submission) and its caption. We also added indications as to where to find all clade names in the revised Results section (see lines 108-114).

- *Lines 121-129: the authors mention the identification of eight points of speciation increase coinciding exactly/one node above points of increase in tooth complexity or plant consumption. However, comparing figs. 2, 4a,b, (points of increasing tooth complexity) with fig. 4c (points of speciation rate shifts), those do not coincide. High speciation is depicted in early squamates, scincids, lacertids, derived snakes, Polyglyphanodontidae, Mosasauria, varanids, and stem iguanians, none of which coincide with points of increase in tooth complexity. Even the points of detected increased speciation within iguanians occur further crownward in the tree when compared to the point of increased tooth complexity in iguanian evolution (with the exception of agamids).*

As noted above, a column of Extended Data Table 3 of the initial submission (Supplementary Table 12 of the current revision of the manuscript) was accidentally cropped out, namely the one detailing dietary transitions (if any) at nodes associated with shifts in speciation. For clarity, we compared the locations of shifts in speciation rate with those of any increase in tooth complexity or diet, not just the six major clades showing increased cusp number highlighted in Fig. 2, including cusp number increase within already multicuspid teeth and increases in proportion of plants consumed within already plant-based diets. We have detailed further the process in the Methods section on diversification models (lines 568-570). One important note is that, while BAMM infers shifts along branches, we chose to reconstruct tooth complexity and

diet at nodes. In order to compare shifts and character transitions, we thus assigned each speciation shift to the node immediately above it. Following these criteria, there are five direct correspondences between increased speciation and increased cusp number or plant consumption (namely, Egerniinae, noted B in Fig. 4c, Polyglyphanodontia, noted C, the genus *Podarcis*, noted D, total group Pleurodonta, noted I, and a sub-clade of Iguanidae, noted M). Furthermore, we identified two correspondences between increased speciation and decreased cusp number (genus *Phrynosoma*, noted K in Fig. 4c, and total group *Gallotia*, which is not displayed in Fig. 4c as this shift was recovered in less than half of our BAMM replicates). For clarity, all direct correspondences with shifts repeated in more than half of our replicates are now indicated via a colour code in Fig. 4c and named in the figure caption. Additionally, we provide the clade names of all shift correspondences (regardless of the number of replicates they appeared in) in the revised text (see lines 161-167). We thus maintain the correspondences we mentioned, which are also detailed in Supplementary Tables 12 and 13 (Extended Data Table 3 and Supplementary Table 6 of the initial submission).

- *As a general challenge to the authors' conclusions, and that they should address, is the well-known fact that some of those lineages with high diversity, such as varanids and derived snakes (i.e. colubroids), have some of the simplest dental shapes among squamates. The Varanus and colubroid radiations represent the most successful taxonomic radiations in squamate evolution, along with the Anolis radiation within iguanians—e.g. (Pianka & Vitt, 2003), clearly indicating that changes in diet and tooth complexity do not relate to their ecological and evolutionary success.*

The reviewer is right to point to highly diverse groups of squamates with single-cuspid teeth. We have added a mention to these clades within the revised discussion (lines 257-260). There is previous research suggesting specific unicuspid dental morphologies or characters and predatory diets may have played a role in the diversification of these groups (Savitzky 1980; Alamillo 2010; Westeen *et al.* 2020), however, we chose to focus our study on the evolution of plant consumption and multicuspid teeth. Besides, in our diversification models we retrieve only *Varanus* as a recurrent shift location towards increased speciation. This suggests that, in contrast, colubroids did not accumulate diversity at exceptional rates for crown Alethinophidia.

- *Lines 131-135: what are those unobserved factors that may have contributed to diversification? You mentioned here “(Extended Data Table 4 and Figure 5)”, but those do*

not make explicit mention of additional factors. Also I believe he referred to Supplementary data figure 5, as there is no Figure 5.

We may have written this sentence too ambiguously regarding the materials we were referring to and their contents. In the brackets, we were calling for Extended Data Table 4 and Extended Data Fig. 5. These are now Supplementary Table 14 and Supplementary Fig. 12 in the revised manuscript; we changed the text and call them earlier in the sentence to make the figure we refer to more explicit (see lines 176-180 of the revised text). This was to point the reader towards the detailed results of our trait-dependent diversification models (relative model support in Extended Data Table 4 and trait-dependent speciation and extinction rates for the best-supported model in Extended Data Fig. 5), rather than specifically to detail diversification factors beyond the scope of our study. Importantly, while HiSSE models explicitly account for a “hidden” unobserved trait, it serves only as a tool to evaluate the relative impact of the focal trait and is unknowable by design (Beaulieu & O’Meara 2016). In detail, factors previously identified as influential for squamate diversification include global temperatures (Garcia-Porta *et al.* 2019; Condamine *et al.* 2019), latitude (Pyron 2014), and habitat (Ricklefs *et al.* 2007, Bars-Clozel *et al.* 2017).

- *Lines 157-159: only two clades are listed among the ones where the authors find a strong support for correlated evolution of dental shape and herbivory (also associated with high speciation). As mentioned elsewhere here, please list all of those clades somewhere in the main text. This is one of the key findings of this manuscript and is key to illustrate their findings and for all readers interested in squamate evolution.*

Here we meant, first, to point out the evidence for correlated evolution of tooth complexity and plant consumption in Squamata as a whole, and second, that increasing tooth complexity and plant consumption appear to have favoured higher speciation in individual squamate groups (note that trait-dependent models of diversification also support increased speciation in multicuspid and plant-consuming species for Squamata as a whole). We have clarified and divided that sentence, and all clade names are now provided in the revised text (see lines 223-227).

- *Lines 157-165: the authors find increasing levels of speciation rates (and net diversification) in squamates between the Early Jurassic and the end of the Cretaceous, with only one visible drop in those levels at around 125Ma during the Early Cretaceous. Those results are interesting, but come into strong conflict with the measures of species richness through time*

using data from the fossil record published over the last two years that suggest low levels of squamate diversity during most of the Mesozoic and a considerable increase in the squamate diversity only at the Late Cretaceous (Cleary et al., 2018; Close et al., 2020; Close et al., 2019). The authors results here are derived from fundamentally different methodologies, but given the fundamental differences between their results in previous ones (with a considerable impact to our understanding of squamate evolution in deep time) I strongly suggest the authors to address such differences in their main text, and potential reasons for those differences.

My impression is that the poor fossil record of squamates from the early Triassic to the late Cretaceous (owing to low research effort, geographical biases in fossil collection, etc) creates biases in diversification dynamics estimated using fossil data with the methods employed in the studies cited above (e.g., subsample diversity data quorum). Perhaps alternative methods with stronger corrections for fossil biases (BAMM and PyRate), such as employed by the authors, are more capable to avoid biases introduced by the poor early fossil record of squamates. However, see also my comments below on the methodology section related to BAMM and phylogeny-based estimates of diversification.

The referee is right to point differences with these previous studies of past squamate diversity. The key differences of these to our own work are:

- i. We considered the rates of speciation and extinction, rather than diversity in terms of the number of species per time bins as shown in these previous studies. These measures relate to the number of branching events per unit of time necessary to recover the observed number of species and known fraction of unsampled species. Since multiple patterns of speciation and extinction can result in the same diversity curve, it is not too unexpected that differences are seen.
- ii. Cleary *et al.* (Cleary *et al.* 2018) consider generic rather than specific diversity, while Close *et al.* estimate specific diversity at the local (Close *et al.* 2019) and regional levels (Close *et al.* 2020).

We feel that discrepancies in the recovered diversification dynamics are to be expected based on such methodological differences. Moreover, there are known issues with both tree-based methods (e.g., extinction rates are notoriously challenging to infer) and estimates from time series (e.g., the duration of time-bins and, as you mention, the quality of the fossil record directly impact the estimations). Therefore, methodological differences of our tree-based approach could explain the exponential increase in speciation and low extinction we observe through the Cenozoic, as opposed to the relatively stable diversity estimates of previous studies

(Cleary *et al.* 2018; Close *et al.* 2019, 2020). We do find however that our Mesozoic pattern of diversification is in broad agreement with these earlier works, in that it features a (local or absolute) diversity peak in the second half of the Cretaceous (specifically in the Campanian) and a post K-Pg increase in diversity (see our new plot of the number of lineages through time in our phylogeny, Supplementary Fig. 11 and our revised discussion lines 227-232). Interestingly, this mirrors patterns recently identified through a time-series approach in non-avian dinosaurs (Condamine *et al.* 2021), which also show a Campanian diversity peak and a Maastrichtian decline in speciation rates (most notably in herbivorous clades), although extinction rates rose over that period rather than decline as in our findings for squamates (see revised discussion lines 239-244, Supplementary Figs. 10-11).

- *Line 163/figure 4d: “net turnover”. I believe you referring here to net speciation (speciation – extinction). “Turnover”, or relative extinction, is something different.*

The figure, its caption, and related text passage referred to turnover as defined by Beaulieu & O’Meara (Beaulieu & O’Meara 2016) – *i.e.*, the sum of speciation and extinction – rather than net speciation/diversification. After concertation between co-authors, we decided to instead use a more standard definition of turnover as the ratio of extinction and speciation rates, now explained in the revised text (lines 171 and 240-241) and displayed in Fig. 4d and Supplementary Fig. 10.

FIGURES AND TABLES

- *Figure 4c: please list the clade names for the clades with considerable speciation rate shifts (A-M) in the figure legend. Alternatively, least those 13 clade names on the side of figure 4c itself. Since nature communications allow up to 10 figures, you can easily split figure 4 so that component 4c is its own figure with enough space available to list those clades.*

I noticed those clades are numbered in Extended Data Table 4, but I would also suggest replacing those numbers for clade names in the table, as the numbers are hard to track without a translation table for clade names.

Done. We have now listed the clades A through M in the captions of Fig. 4 and Supplementary Table 12 and 13 (the former Extended Data Table 3 and Supplementary Table 6, respectively). Note that we limited names to the captions due to several clades not having a formal name (to the best of our knowledge) and that need to be referred to by the definition of a most recent common ancestor to extant species.

- *Figure 4d. This is a key figure where the authors present the results of their speciation and extinction rates through time. Yet, this is a very small component of what is already a figure component. I suggest adding at least a much larger version of this figure as a supplementary figure so readers can see this figure in much greater detail, or a new figure of its own in the main text.*

Done. We have added a larger version of panel 4d as Supplementary Fig. 10. A smaller version remains in the main text to be displayed together with panel 4c.

- *In the supplementary information, tables are provided to indicate the branches of the tree where dietary and tooth complexity transitions are happening, but the authors do not provide a figure with the nodes plotted in the tree so the readers can track down those node numbers in the tree. I strongly recommend adding that to the supplementary information, even if you need to break up the tree across three or four separate figures (and pages) because of the size of the tree. This makes it much easier for readers to directly assess the particular clades where those transitions are happening, without having to download the nexus tree files, import them into R, etc.*

Done. We have now mapped all node and branch numbers onto the polytomous and dichotomous versions of our phylogeny as Supplementary Figs. 1-4.

METHODOLOGY

- *Phylogenetic supertree (full species tree provided in the supplementary information): Different squamate systematists will definitely have their own preferences over specific aspects of the tree, an issue that is always impossible to avoid. However, I find that the overall topology and calibration times for their tree looks quite reasonable and I believe most systematists would agree with the overall tree structure. However, there is one aspect of the tree that I find it necessary to point out as it will be highly controversial for anyone familiar with snake evolution: the paraphyly of simoliophiids. Every squamate phylogeny including fossil snakes that I am aware of including two or more species of simoliophiids always recovers simoliophiids as monophyletic. Currently, the authors place Haasiophis+Pachyophis in a distinct location from Pachyrhachis, further up in the tree. Although the placement of simoliophiids in snake phylogenies is constantly shifting, one thing everybody agrees on is on their monophyly (wherever they may be in the phylogeny of snakes). Having said that, the fact that all snakes have pretty much the same dental morphology will not affect their results on the rates of*

morphological evolution. But since this is a very controversial reconstruction of early snake phylogeny, I strongly recommend the authors to, at least, address this issue in their methods. We followed Simões *et al.* (Simões *et al.* 2018) for the relationships of *Najash* and *Pachyrhachis*, and Pyron (Pyron 2016) for the placement of *Haasiophis* and *Eupodophis* (though in the latter *Pachyrhachis* is indeed considered a simoliophiid). Unfortunately, Simões *et al.* only considered *Pachyrhachis* in their study, and we reflected both possible placements for simoliophiid snakes in our tree. Also, none of these species were identified as Simoliophiidae in the Paleobiology Database, which we used as our reference for the taxonomy of fossil squamates. As suggested by the reviewer, the placement of these fossils is now justified in the revised Methods (lines 366-368). In agreement with the reviewer's observation, we do not expect such variations in snake phylogeny to affect the results of our study.

- *Geometric morphometrics: the PCA analysis and morphospace plotting (Fig. 1c) is important to demonstrate the overall distribution of variance in the data, but an actual discriminant analysis between groups (CVA or between group PCA) would be a better approach to demonstrate significant separation between the groups.*

We demonstrated a significant separation of the groups using phylogenetic multivariate analysis of variance (MANOVA; see lines 96-99 and Supplementary Table 2). We thus found all groupings to be significantly mutually distinct (except omnivores from predators; see Supplementary Table 2) with negligible phylogenetic signal ($\lambda = 0.03$). In addition, we have now computed a discriminant function analysis (DFA) on the regularized variance-covariance matrix of the fitted Pagel's lambda model used for the MANOVA to emphasise the separation between groups (see lines 101-103, Supplementary Fig. 5). However, it is important to note that a MANOVA is nevertheless necessary to confirm the statistical significance of the groupings chosen for the DFA.

- *Tests of correlated evolution (lines 450-455): for the analysis with BayesTraits, in a similar manner to the BAMM analysis below, authors should explicitly specify parameters for their Bayesian analysis. The authors should indicate the number of independent runs, number of chains in each run, and what are the diagnostic reports for convergency and stationarity.*

Done; we detail these settings and parameters in the revised Methods (lines 483-487). We used two independent runs (ten for diversification models with BAMM), each with four independent chains. In all cases, we assessed convergence and stationarity through visual examination of the traces and checked for large effective sample size with coda for R.

- *Models of Diversification (lines 505-525): the authors utilized BAMM v. 2.6 for their inferences on the shifts of speciation and extinction rates. As the authors correctly indicate, this is the most complete method to infer such rates from phylogenetic trees, as it includes the assumption of fossil sampling and fossilization rates. However, the authors do not indicate why they chose a phylogeny-based approach to estimate rates of speciation and extinction to begin with. Competing approaches, especially the latest version of PyRate (Silvestro et al., 2019), also provide estimates of speciation and extinction that use the vast amount of fossil record data available from the Paleobiology Database, and which are independent of potential variations in phylogenetic tree shape. It is not yet clear which of these two approaches is more accurate, but it would be definitely important for the authors to lay out their rationale for choosing BAMM to estimate speciation and extinction rates as opposed to other approaches, especially PyRate.*

As the reviewer pointed out, we used BAMM 2.6 because it is currently the only approach available allowing the estimation of branch-specific diversification rates on phylogenetic trees that include fossils (non-ultrametric trees). Recent and more robust approaches such as ClaDS (Maliot *et al.* 2019) cannot handle non-ultrametric trees yet.

Our choice for a tree-based estimate is twofold:

- i. This allows us direct comparison with Maximum Likelihood ancestral trait reconstructions and branch-specific phenotypic rate estimates obtained from BayesTraits (which are based on a similar rjMCMC implementation).
- ii. While both approaches – time series based on occurrence number and phylogenetic approaches based on a splitting process – have their own merits and limits (*e.g.*, accuracy of the tree, fossil record, sampling biases) they may capture different aspects. Tree-based rate estimates allow to assess variations in speciation rate not only through time, but also across lineages. This removes ambiguity in defining time series for different clades and copes at the same time with the poor fossil record of the studied groups.

Our reasoning is now detailed in the revised Methods (lines 543-552).

- *One specific point I believe authors should address is whether they believe that their rates of extinction may be affected by using a phylogeny-based approach. This is a common problem for extant phylogenies based on molecular data (Quental & Marshall, 2010; Silvestro et al., 2018), which in theory, the authors may have circumvent by using fossil data*

and BAMM v 2.6. Yet, the results presented here seem to have exceptionally low levels of extinction between 250-150Ma. If this extinction level is being underestimated, this could explain the steady increase in net diversification in squamates during the Jurassic, which contrasts to the results presented by recent studies using the fossil record (see also comments above for Lines 157-165).

The reviewer is right; estimating extinction rate from phylogenies of extant taxa is a difficult task (Rabosky 2010), which is why we used a tree including both extinct and extant species in our analysis. To the best of our knowledge, this is the first attempt at such a study of squamate evolution. Although potential biases remain in the estimates of extinction rates, we think that the relative changes in turnover are indicative of changes in diversification dynamic. The K-Pg boundary marks the only period previously reported to have a considerable impact on squamate extinction (Longrich *et al.* 2012). Though we did not retrieve signal for increased extinction, the boundary marks a clear phase shift in the diversification dynamics we recover, which overall does not affect our interpretation. We now reflect this comment in the main text (lines 227-232).

- *For their implementation of BAMM, it is very important for the authors to provide some additional details on their implementation. For instance, they indicate they used priors generated by BAMMtools, but they do not specify what are, specifically, those prior choices. Additionally, they choose a preservation rate of 0.01 for the squamate fossil record, but do not indicate how they came up with this particular value. Please, elaborate more on those aspects as they are crucial to BAMM results.*

The relevant Methods section of the revised manuscript includes the prior values determined by `setBAMMpriors` (namely `expectedNumberOfShifts = 1.0`; `lambdaInitPrior = 10.2511191939331`; `lambdaShiftPrior = 0.00400165322948176`; `muInitPrior = 10.2511191939331`; see lines 557-559). Because we could find no published or reported estimate of preservation rate for Squamata, we arbitrarily chose a preservation rate prior of 0.01 to reflect the poor quality of their fossil record (Cleary *et al.* 2018). In preliminary runs, changes of several orders of magnitude in the prior yielded negligible differences in the results; moreover, the authors of the method have shown its robustness to heterogeneity in preservation rates (Mitchell *et al.* 2018). We amended the Methods section to make explicit that our choice was made in the absence of a published proposal for this parameter to the best of our knowledge (lines 555-557).

- *Statistics (line 544): ...” Macroevolutionary rates...”*: Please replace that by *speciation and extinction rates. Macroevolutionary rates, or rates of evolution, can be used for either diversification dynamics as well as rates of character evolution.*

Done. We made corrections to mention “speciation and extinction rates” instead.

Replies to Reviewer #2:

- *The authors present a study that describes the evolutionary history of dentition in reptiles. They present several lines of evidence showing that complexity in dentition is related to diet and has gone through transitions across the reptile lineage. Some of these transitions include evolving more complex or losing complexity in teeth morphology (namely, cusps). They postulate that changes in dentition and diet also contribute to the diversification of the reptile lineage.*
- *In my opinion, the study is strong, analyses are appropriate, and their claims are validated via several lines of evidence. I’m incredibly impressed by the study and I failed to find any large issues that would prevent the study from being published. The PCMs used are what is standard in the field and their interpretations are reasonable, which is refreshing as many studies I have reviewed in the past have a hard time with PCMs and what they mean.*
- *My one criticism of the study, and this is minor, is that I would like to see the authors explicitly test the hypothesis that dentition complexity is related to diet using an OU method (SLOUCH, for example) and present the adaptive landscape to provide further evidence that there is an adaptive optima for each kind of dentition and that the adaptive landscape is shaped by diet. This should be relatively quick to do and only add a few lines to the manuscript but would ease my concern that there are real peaks and valleys in the landscape and that dentition complexity is near an adaptive peak.*

We thank the reviewer for this suggestion. In the revised manuscript, we fitted several variants of multivariate models of continuous trait evolution (Brownian Motion (BM), Early Burst (EB), and Ornstein-Uhlenbeck (OU)) on the first five PC axes describing the tooth outline dataset (see Supplementary Tables 10, 11 and Supplementary Fig. 9). The results show overwhelming support for a multi-peak Ornstein-Uhlenbeck model with clearly separated optima corresponding to each diet, as now described in the revised text (lines 150-153). Nevertheless, we want to point out that, depending on the evolution of the adaptive landscape, adaptation to different ecological strategies does not necessarily follow an OU process.

- *Otherwise, I commend the authors on the tremendous work they put into the study and believe this study will be of great value to the scientific community.*

We thank the reviewer for their encouraging words and commendation.

Replies to Reviewer #3:

SUMMARY

- *Lafuma et al. reconstruct patterns of tooth evolution in Squamata, one of the most diverse tetrapod clades, and compare these patterns to Mammalia. Insofar as tooth evolution in Mammalia is thought to be an important part of the success of the clade, studying a parallel radiation and comparing and contrasting the dynamics is intriguing, and they find distinct macroevolutionary dynamics (higher rates of complexity acquisition- and loss- but it seems less complexity than achieved by mammals). The paper is well written and seems solid and the conclusions are of broad general interest to people interested in macroevolution, rather than just lizard evolution, and therefore appropriate for publication for a high profile journal such as Nature Communications. In my opinion it's publishable with minor revisions although I'm more a morphology guy than a stats/tree guy, so others might have thoughts on the methods.*

We thank the reviewer for their kind words and especially for the very interesting discussion points below, which we hope to have replied to in a similar spirit!

- *I think my one suggestion would be that it might be better to try to somehow map out these evolutionary events over time, perhaps by (1) plotting the changes in cusp number (increases and decreases in cusp count) over geological time, and/or (2) trying to look at tooth disparity (even just simple variables like crown width and cusp count) over time. I think doing this would help better communicate the results and the story. Some photos of representative tooth morphologies (as it stands, the emphasis on trees and plots makes it feel very abstract) might help provide context to the reader, as well as a bit more discussion of the kinds of morphologies seen in squamate teeth (which are more diverse than one might think). These are suggestions and my recommendation to accept isn't contingent on them.*

We thank the reviewer for their helpful suggestions. We have added a time-binned plot of all levels of tooth complexity and diet through time according to the results of our ancestral state reconstructions as a complement to Fig. 3 (panels a and c). We also discuss in further detail the

dental morphospace issued from our morphometric analysis (see lines 89-96) and the more subtle variations of squamate multicuspid tooth morphologies (see lines 207-218).

DETAILED COMMENTS

- *I think a little more detail on how squamates achieved multicuspid teeth might be useful, perhaps even figures or photographs showing the morphology. Just counting cusps doesn't really communicate the variety of tooth morphologies in lizards; there's more going on than one might think at first. For example, the polyglyphanodonts achieve tooth complexity several different ways. Polyglyphanodon expands the crown transversely to create transverse blades (this may or may not be related to the weird multicuspid arrangement seen in Peneteius). However, Macrocephalosaurus achieves tooth complexity by expanding teeth anteroposteriorly and evolving many small cusps, like a modern iguana. I think a nice summary figure- photographs showing the range of tooth morphologies seen in squamates- would also help communicate this? Some discussion might be useful as well.*

These are very relevant points in both fossil and living squamates that we have reflected in our extended discussion (lines 207-218).

- *Also, dolichosaurs, specifically Coniasaurus, have weird teeth and arguably you could call it bicuspid (the anterior carina is strongly convex in Coniasaurus) so that may be another evolution of multicuspid teeth?*

We thank the reviewer for pointing this multicuspid species, which we had not previously identified. Upon inspection, *Coniasaurus* specimens in the literature do present at least some teeth that can be considered bicuspid. We have not repeated our analyses with the inclusion of *Coniasaurus* but will make sure to include it in future further studies. This is a good confirmation that we did not sample all independent originations of complex teeth and that the number of originations we report is most likely a minimum estimate. We now mention this aspect in the revised Discussion (lines 191-192).

- *“we reconstruct the MRCA of Squamata as unicuspid and infer at least 24 independent acquisitions of multicuspidness in squamate lineages.”*
Sorta burying the lead here! This is really striking. Even if squamates never achieved the complexity seen in mammals, they evolve multicuspid teeth far more often, it seems, and in that sense they're very innovative. This is a really striking result and I think should be up in the abstract.

We now reference more explicitly this result in the abstract (line 22-23).

- *“while resulting in more labile dental complexity throughout evolution.”*

Well it's like mammals didn't do it often, but did it really well, versus lizards which did it often but never quite to the extreme seen in mammals; even the most elaborate lizard teeth arguably aren't as complex as typical mammal dentition. So perhaps lizards were more evolutionarily versatile, but mammals achieved higher complexity. Maybe the two-complexity and versatility- are difficult to reconcile? We can argue about which is better. There are more lizard species, but mammals have higher biomass, probably occupy more niches (e.g. whales, bats, elephants have no squamate competitors).

While we agree that an evolutionary trade-off between the level of tooth complexity and its evolvability may exist, recent research also suggests the development of complex morphologies is qualitatively different from simple ones. Namely, increasingly complex morphologies need to be finely tuned through more complex genotype-phenotype maps, as mutations in their underlying gene regulatory networks become more and more likely to decrease complexity (Hagolani *et al.* 2021). Experimental data in mammals (Harjunmaa *et al.* 2012) and the developmental characteristics of squamate teeth we already mentioned in our discussion (lines 272-275) fit this idea of simpler, labile squamate morphologies as opposed to mammals' complex “developmentally locked” teeth. We have added the first reference to the discussion, although we did not develop further this argument so as to not too heavily focus on mammals rather than squamates.

The reviewer is also right to mention that, in modern environments, mammals occupy more niches than squamates. As a slight caveat, we would like to point out that, while squamates lack active flyers, gliding species exist, and, like many bats, most squamate exploit insect resources, possibly reaching their prey through living in the canopy rather than active flight. We added comments on the Cenozoic ecological diversification of mammals and squamates to the discussion (lines 200-203 and 248-252).

- *Archosaurs seem sort of intermediate? The teeth of ceratopsians, hadrosaurids etc. are more complex than those of lizards arguably, less so than mammals in my opinion. They didn't evolve complex teeth all that often- there's a single clade, Ornithischia, that has most of the complex teeth of the entire group, with some tooth complexity in Notosuchia. But arguably at some level how often you evolve complexity is less relevant than how well you do it? The eukaryotic cell and oxygenic photosynthesis are one-off innovations (like the*

tribosphenic molar) but they revolutionized life on Earth; they didn't evolve often but it made a huge difference. Humans (social animals with sophisticated tool use and language and giant brains) are a one-off event but have completely reshaped the biosphere.

We agree with the reviewer and added these aspects to our discussion (lines 199-207).

- *“mammal teeth are more integrated structures, less prone, through intense selective pressures, to loss of complexity”*

*That's an interesting problem. I guess the question is, are they intrinsically less evolvable due to developmental and anatomical constraints, or just exposed to different selective pressures? You do get homodont, unicuspid teeth in certain lineages- odontocete whales, armadillos. Numbats (*Myrmecobius*) do a weird thing where the teeth are functionally homodont/unicuspid like a lizard, but this is achieved by aligning the three cusps of the molars to form the functional equivalent of three, unicuspid teeth... which seems to suggest that mammals do struggle to evolve unicuspid teeth, even faced with the strong selective pressures for an insectivorous diet, they're cheating by making a functionally unicuspid arrangement out of multicusped teeth. Just as often mammals seem to lose the teeth entirely (pangolins, anteaters, various cetaceans) which offhand I don't think squamates ever did?*

We agree with the reviewer and thank them for the additional examples they provided. Experimental data support the idea that, in mammals, developmental changes lead more often to complete tooth loss than reduction in cusp number (Harjunmaa *et al.* 2012, 2014), which may make it difficult to reach unicuspid morphologies even under appropriate selective pressures (see also, *e.g.*, seals (Savriama *et al.* 2018)). Note that, as this consideration centres around mammals, we have not added it to the discussion to preserve our focus on squamates. To the best of our knowledge, complete tooth loss has not been documented in any living or fossil squamate, though some scolecophidian and colubrid snakes show significant reduction of the dentition (Gans 1952, Rieppel *et al.* 2009). We have included this interesting point in the discussion (line 194).

- *To me this suggests a tradeoff between complexity and evolvability. Mammals have achieved extremely high levels of tooth complexity, but perhaps the various developmental pathways needed to make a placental molar now limit the directions you can take it?*

We fully agree with the reviewer's suggestion; see replies above.

- *Re: the K-Pg boundary it's a little surprising to see that more doesn't happen there. In part, I think it's extremely difficult to time some of these radiations and better models and calibrations may yet push some of these radiations into the Cenozoic (e.g. crown Pleurodonta) but even so, this wouldn't affect things a lot. That is, even if crown Pleurodonta is a Paleogene radiation, they already had multicuspid teeth in the Cretaceous, and we don't see a lot of acquisition in the basal branches of that tree.*

This does sort of fit with my sense that lizards failed to innovate the way mammals do in the Cenozoic. Many mammal lineages shift from omnivory to herbivory, but lizards seem far more conservative in the Cenozoic- arguably they less innovative in the Cenozoic than in the Cretaceous where we had for example, mosasaurs, various polyglyphanodont lineages evolving radically different tooth morphologies.

We concur with the reviewer in finding that squamates saw more morphological and ecological innovations during the Mesozoic than the Cenozoic, which is in line with the recent findings of Simões *et al.* pointing to a peak in lepidosaur phenotypic disparity and both phenotypic and molecular rates of evolution in the latest Cretaceous (Simões *et al.* 2020). The Cenozoic was nevertheless an important period in term of ecological innovation, as we infer most origins of plant consumption during that period, though without resulting in a diversity of herbivores comparable to mammals. We added these considerations to the discussion (lines 200-203 and 248-252).

- *With respect to the discussion it might be useful to discuss squamates in terms of what else is happening in the Mesozoic, in other clades? This is sort of touched on (the KTR) but note that we also have other groups experimenting with herbivory at this time. E.g. lots of herbivorous theropods- Oviraptorosauria, Ornithomimidae, Therizinosauroidea, even certain Troodontidae- show up in the Cretaceous. Highly specialized, herbivorous multituberculates also diversify, e.g. Meniscoessus. Herbivorous crocodylomorphs, like Simosuchus appear. Herbivorous turtles, e.g. Basilemys. It's a big time for many small herbivores, not just lizards.*

We agree with the reviewer and have included these aspects in the revised discussion (lines 237-239).

- *Minor issues: "do not" rather than "don't" in the abstract ("don't" and other contractions being more informal)*

We have corrected this in our updated abstract.

References

- Alamillo, H. Testing macroevolutionary hypotheses: diversification and phylogenetic implications. (Washington State University, Pullman, 2010).
- Bars-Closel, M., Kohlsdorf, T., Moen, D. S., & Wiens, J. J. Diversification rates are more strongly related to microhabitat than climate in squamate reptiles (lizards and snakes). *Evolution* **71**, 2243-2261 (2017)
- Beaulieu, J. M. & O'Meara, B. C. Detecting hidden diversification shifts in models of trait-dependent speciation and extinction. *Syst. Biol.* **65**, 583-601 (2016).
- Cleary, T. J., Benson, R. B., Evans, S. E. & Barrett, P. M. Lepidosaurian diversity in the Mesozoic–Palaeogene: the potential roles of sampling biases and environmental drivers. *R. Soc. Open Sci.* **5**, 171830 <https://doi.org/10.1098/rsos.171830> (2018).
- Close, R. A. *et al.* Diversity dynamics of Phanerozoic terrestrial tetrapods at the local-community scale. *Nat. Ecol. Evol.* **3**, 590-597 (2019).
- Close, R. A. *et al.* The apparent exponential radiation of Phanerozoic land vertebrates is an artefact of spatial sampling biases. *Proc. R. Soc. B* **287**, 20200372 <https://doi.org/10.1098/rspb.2020.0372> (2020).
- Condamine, F. L., Guinot, G., Benton, M. J., & Currie, P. J. Dinosaur biodiversity declined well before the asteroid impact, influenced by ecological and environmental pressures. *Nat. Commun.* **12**, 3833 <https://doi.org/10.1038/s41467-021-23754-0> (2021).
- Condamine, F. L., Rolland, J., & Morlon, H. Assessing the causes of diversification slowdowns: temperature-dependent and diversity-dependent models receive equivalent support', *Ecol. Lett.* **22**, 1900-1912 (2019).
- Gans, C. The functional morphology of the egg-eating adaptations in the snake genus *Dasyeltis*. *Zoologica (N. Y.)* **37**, 209-244 (1952).
- Garcia-Porta, J. *et al.* Environmental temperatures shape thermal physiology as well as diversification and genome-wide substitution rates in lizards. *Nat. Commun.* **10**, 4077 <https://doi.org/10.1038/s41467-019-11943-x> (2019).
- Hagolani, P. F., Zimm, R., Vroomans, R., & Salazar-Ciudad, I. On the evolution and development of morphological complexity: A view from gene regulatory networks. *PLoS Comput. Biol.* **17**, e1008570 <https://doi.org/10.1371/journal.pcbi.1008570> (2021).
- Harjunmaa, E. *et al.* On the difficulty of increasing dental complexity. *Nature* **483**, 324-327 (2012).
- Harjunmaa, E. *et al.* Replaying evolutionary transitions from the dental fossil record. *Nature* **512**, 44-48 (2014).
- Longrich, N. R., Bhullar, B. A. S., & Gauthier, J. A. Mass extinction of lizards and snakes at the Cretaceous-Paleogene boundary. *Proc. Natl Acad. Sci. USA* **109**, 21396-21401 (2012).
- Maliot, O., Hartig, F., & Morlon, H. A model with many small shifts for estimating species-specific diversification rates. *Nat. Ecol. Evol.* **3**, 1086-1092 (2019).
- Mitchell, J. S., Etienne, R. S. & Rabosky, D. L. Inferring diversification rate variation from phylogenies with fossils. *Syst. Biol.* **68**, 1-18 (2018).
- Pyron, R. A. Temperate extinction in squamate reptiles and the roots of latitudinal diversity gradients. *Glob. Ecol. Biogeogr.* **23**, 1126-1134 (2014)

- Pyron, R. A. Novel approaches for phylogenetic inference from morphological data and total-evidence dating in squamate reptiles (lizards, snakes, and amphisbaenians). *Syst. Biol.* **66**, 38-56 (2016).
- Rabosky, D. L. Extinction rates should not be estimated from molecular phylogenies. *Evolution* **64**, 1816-1824 (2010).
- Ricklefs, R. E., Losos, J. B., & Townsend, T. M. Evolutionary diversification of clades of squamate reptiles. *J. Evol. Biol.* **20**, 1751-1762 (2007).
- Rieppel, O., Kley, N. J., & Maisano, J. A. Morphology of the skull of the white-nosed blindsnake, *Liotyphlops albirostris* (Scolophorida: Anomalepididae). *J. Morphol.* **270**, 536-557 (2009)
- Savitzky, A. H. The role of venom delivery strategies in snake evolution. *Evolution* **34**, 1194-1204 (1980).
- Savriama, Y. *et al.* Bracketing phenogenotypic limits of mammalian hybridization. *R. Soc. Open Sci.* **5**, 180903 <https://doi.org/10.1098/rsos.180903> (2018).
- Silvestro, D., Salamin, N., Antonelli, A., & Meyer, X. Improved estimation of macroevolutionary rates from fossil data using a Bayesian framework. *Paleobiology* **45**, 546-570 (2019).
- Simões, T. R. *et al.* The origin of squamates revealed by a Middle Triassic lizard from the Italian Alps. *Nature* **557**, 706-709 (2018).
- Simões, T. R., Vernygora, O., Caldwell, M. W., & Pierce, S. E. Megaevolutionary dynamics and the timing of evolutionary innovation in reptiles. *Nat. Commun.* **11**, 3322 <https://doi.org/10.1038/s41467-020-17190-9> (2020).
- Westeen, E. P., Durso, A. M., Grundler, M. C., Rabosky, D. L., & Rabosky, A. R. D. What makes a fang? Phylogenetic and ecological controls on tooth evolution in rear-fanged snakes. *BMC Evol. Biol.* **20**, 80 <https://doi.org/10.1186/s12862-020-01645-0> (2020).

REVIEWER COMMENTS

Reviewer #1 (Remarks to the Author):

I appreciate the detailed response from the authors for all the queries raised by myself and other referees. I find that the manuscript has indeed included several suggestions and it is much improved on the quality of images, clarity, and especially by the better detailed explanation of results in the results and discussion sections. As mentioned in my previous review, I find this work of great interest to squamate evolution and evolutionary biologists interested on the drivers of modern biodiversity. I congratulate the authors on their ambitious task and careful methodological procedures.

I still have a few, mostly minor, points to address, which I detail below:

Major comment: Snakes and Varanus as an exception to the broader squamate pattern:

First round comments: “As a general challenge to the authors’ conclusions, and that they should address, is the well-known fact that some of those lineages with high diversity, such as varanids and derived snakes (i.e. colubroids), have some of the simplest dental shapes among squamates. The Varanus and colubroid radiations represent the most successful taxonomic radiations in squamate evolution, along with the Anolis radiation within iguanians—e.g. (Pianka & Vitt, 2003), clearly indicating that changes in diet and tooth complexity do not relate to their ecological and evolutionary success.”

Authors’ response: “The reviewer is right to point to highly diverse groups of squamates with single-cusped teeth. We have added a mention to these clades within the revised discussion (lines 257-260). There is previous research suggesting specific unicuspid dental morphologies or characters and predatory diets may have played a role in the diversification of these groups (Savitzky 1980; Alamillo 2010; Westeen et al. 2020), however, we chose to focus our study on the evolution of plant consumption and multicuspid teeth. Besides, in our diversification models we retrieve only Varanus as a recurrent shift location towards increased speciation. This suggests that, in contrast, colubroids did not accumulate diversity at exceptional rates for crown Alethinophidia.”

Second round comments: I appreciate the efforts made by the authors to clarify some of those aspects. As in the previous version of this MS it wasn’t clear the exact clade names representing nodes undergoing rate shifts, I made reference to clade names (varanids and colubroids) that herpetologists know to be abundantly diverse and appeared from the figures to represent the clades undergoing major shifts in speciation rates. As the authors now make it clearer in their new supplementary tables and fig. 4, Varanus (node H, with hundreds of species and representing most of extant varanids), and crown

alethinophidians (node F, representing ~3000 species of extant snakes), and the snake genus *Chilabothrus* (node G) all undergo increases in speciation rates according to the authors results despite maintaining a simple dental morphology. Therefore, my initial comment sustains.

The authors highlight a lot of the clades for which they found concomitant increases in speciation rates and increases in dental complexity or plant consumption in their results and discussions (e.g., Lines 159-181). Yet, two important squamate radiations (crown alethinophidians and *Varanus*) experience increases in speciation rates although retaining a single cusp dentition (i.e., simple dentition under the authors own definition of dental complexity), besides being mostly carnivorous. It is really important to acknowledge such important exceptions in the text more clearly. They do make mention of snakes at the end of their discussion (lines 257-260), but only to mention that they make other kinds of dental- dietary adaptations (such as the development of venom grooves). However, venom grooves are specific only to viperids, thus representing a small fraction of all crown alethinophidians, and venom grooves are not the criterion of complexity used by the authors nor the morphological variable used in their analyses. Importantly, the pattern in crown alethinophidians and *Varanus* deviates from the data and results that they actually do get: that an increase in dental complexity does not explain the major taxonomic radiation of these two lineages. I would highly suggest the authors to be more straightforward and explicit on this result, as this does not diminish the impact of their results and makes it clear how unique snakes are in terms of macroevolutionary dynamics relative to other groups of squamates.

Minor comments:

Line 63: Squamates first appearance in the fossil record is in the Triassic. The estimated origin of the group is in the Permian. The current sentence is ambiguous, so I would suggest making reference here to the estimated divergence time for squamates (not appearance).

Lines 158-164. The authors make mention of five groups in which speciation increases coincide exactly with increases in tooth complexity or plant consumption: Egerniinae, Polyglyphanodontia, Podarcis, total group Pleurodonta, and a sub-clade of Iguanidae (listed in supplementary table 12). However, Egerniinae (node B) and Podarcis (node D), listed above, do not have any transition in cusp number and the data transition is from insectivorous to omnivorous. This may sound picky but most readers when reading increasing plant consumption may assume transition to herbivory. The text could be improved by stating something like "...or increase in plant consumption (transitions into herbivory or from insectivory to omnivory)."

Lines 158-164. The authors have enough room to include the names of the five clades (one node above or below) that also face increases in speciation rates along with tooth complexity/diet. It would be much easier for the reader to have those names here, as the authors did for the other five clades, as most will not have the time or patience to look for them on the supplementary tables. The same with the five

lineages that faced the opposite pattern (increases in speciation concomitant with decreases in tooth complexity/diet).

Lines 170- 171: the authors dropped a comparison between the apex of speciation in squamates with “a critical timing angiosperm evolution” for no apparent reason or connectivity with the rest of the text here. I see the bring some discussion on the evolution of angiosperms and squamates later in the discussion, but the present sentence is out of context. Either rephrase this sentence or simply remove and keep this for the discussion only.

Lines 176-180. I would suggest rephrasing the result to reflect that this explains the evolution of several major components of squamate diversity, but not all of them, most notoriously, crown alethinophidians (one third of all squamate diversity). See other comments above. So authors could write something such as “contributed considerably to the diversification of non-ophidian squamates”.

Discussion: I suggest the authors should add to their discussion the following lines they provided in the response document, as additional factors contributing to diversification dynamics in squamates: “Factors previously identified as influential for squamate diversification include global temperatures (Garcia-Porta et al. 2019; Condamine et al. 2019), latitude (Pyron 2014), and habitat (Ricklefs et al. 2007, Bars-Closel et al. 2017).”

Supplementary Figure 11 | Squamate diversity through time: It’s important to report whether this line were present mean or median estimates across time bins. I find it important to also include the credibility interval around the summary trendline depicted here. However, if you provide net diversification curves (speciation – extinction) from the BAMM results I think it could replace this figure altogether.

Supplementary table 12. You list all the clades for each node number but would be helpful to repeat here in this legend the five clades with coincident increases in tooth complexity or plant consumption that you mention in the main text. You did this for supplementary table 13, which was helpful, and both for consistency and readership it would be informative here as well.

Reviewer #3 (Remarks to the Author):

As previously stated, I feel the manuscript was a strong candidate for inclusion in Current Biology as it stood but felt there were places it could better communicate and explore the results. The plots of tooth complexity over time do help the reader visualize the results more readily. I'm happy to recommend acceptance.

Nature Communications manuscript NCOMMS-20-23722A

Multiple evolutionary origins and losses of tooth complexity in squamates

F. Lafuma, I. J. Corfe, J. Clavel, N. Di-Poi

Point-by-point replies to the reviewers' and editorial comments and suggestions

We thank the two reviewers for the renewed positive evaluation of our manuscript. We have thoroughly considered all their comments and recommendations and detail below how we further revised the manuscript to comply with these observations. We reproduce the referees' comments in italics; our responses and explanations are in plain text.

Replies to Reviewer #1:

I appreciate the detailed response from the authors for all the queries raised by myself and other referees. I find that the manuscript has indeed included several suggestions and it is much improved on the quality of images, clarity, and especially by the better detailed explanation of results in the results and discussion sections. As mentioned in my previous review, I find this work of great interest to squamate evolution and evolutionary biologists interested on the drivers of modern biodiversity. I congratulate the authors on their ambitious task and careful methodological procedures.

We thank the reviewer for their positive assessment and their additional suggestions and comments below, which helped us further improve our manuscript.

I still have a few, mostly minor, points to address, which I detail below:

MAJOR COMMENT: SNAKES AND VARANUS AS AN EXCEPTION TO THE BROADER SQUAMATE PATTERN

- *First round comments: "As a general challenge to the authors' conclusions, and that they should address, is the well-known fact that some of those lineages with high diversity, such as varanids and derived snakes (i.e. colubroids), have some of the simplest dental shapes among squamates. The Varanus and colubroid radiations represent the most successful taxonomic radiations in squamate evolution, along with the Anolis radiation within*

iguanians—e.g. (Pianka & Vitt, 2003), clearly indicating that changes in diet and tooth complexity do not relate to their ecological and evolutionary success.”

- *Authors’ response: “The reviewer is right to point to highly diverse groups of squamates with single-cusped teeth. We have added a mention to these clades within the revised discussion (lines 257-260). There is previous research suggesting specific unicuspid dental morphologies or characters and predatory diets may have played a role in the diversification of these groups (Savitzky 1980; Alamillo 2010; Westeen et al. 2020), however, we chose to focus our study on the evolution of plant consumption and multicuspid teeth. Besides, in our diversification models we retrieve only Varanus as a recurrent shift location towards increased speciation. This suggests that, in contrast, colubroids did not accumulate diversity at exceptional rates for crown Alethinophidia.”*
- *Second round comments: I appreciate the efforts made by the authors to clarify some of those aspects. As in the previous version of this MS it wasn’t clear the exact clade names representing nodes undergoing rate shifts, I made reference to clade names (varanids and colubroids) that herpetologists know to be abundantly diverse and appeared from the figures to represent the clades undergoing major shifts in speciation rates. As the authors now make it clearer in their new supplementary tables and fig. 4, Varanus (node H, with hundreds of species and representing most of extant varanids), and crown alethinophidians (node F, representing ~3000 species of extant snakes), and the snake genus Chilabothrus (node G) all undergo increases in speciation rates according to the authors results despite maintaining a simple dental morphology. Therefore, my initial comment sustains.*

We agree with the reviewer that these three clades exemplify instances of increased speciation within unicuspid predatory groups, without links to increasing tooth complexity or plant consumption. Mosasauria (node E of Fig. 4c) is another such example. Indeed, our results and interpretations do not exclude the possibility for unicuspid and/or predatory squamate groups to undergo significant diversification bursts, as further demonstrated by the significantly stronger support for a HiSSE rather than a BiSSE model of diversification for both tooth complexity and diet (Supplementary Table 14). Conversely, speciation increases in unicuspid predatory groups do not imply that these shifts are unrelated to tooth morphology or diet. By design, the dental-dietary adaptations of these clades lie beyond our scope (*i.e.*, multicuspid teeth and plant-based diets), and thus our study, as it stands, cannot ascertain whether specific aspects of their unicuspid morphologies and/or predatory diets may explain their diversification

(which could be tested, *e.g.*, through different trait-dependent diversification models). Multiple studies point the repeated evolution of various degrees of venom grooves in several independent clades of Colubroidea and corresponding shifts in diet and prey capture mode as key to their Neogene diversification in a context of extending grasslands and diversifying rodents (Savitzky 1980; Alamillo 2010; Westeen *et al.* 2020; Palci *et al.* 2021). Likewise, mosasaurs underwent an adaptive radiation during the Late Cretaceous while evolving a variety of sometimes highly specialised unicuspid morphologies associated with the piercing, cutting, or crushing of different prey items (Massare 1987; Bardet *et al.* 2005, Hornung & Reich 2015, Longrich *et al.* 2021). The diet of monitors is often reliant on vertebrate preys and blade-like ziphodont teeth have been shown to be critical in de-fleshing and prey reduction while also enhancing venom delivery in *Varanus komodoensis* (Losos & Greene 1988; D'Amore & Blumenschine 2009; Fry *et al.* 2009). The reviewer is right to point the importance of these clades in comparison of the general pattern we describe in Squamata, and we have emphasised that aspect in our revised Results (lines 172-174) and Discussion (lines 265-272; see also our comment below).

- *The authors highlight a lot of the clades for which they found concomitant increases in speciation rates and increases in dental complexity or plant consumption in their results and discussions (e.g., Lines 159-181). Yet, two important squamate radiations (crown alethinophidians and Varanus) experience increases in speciation rates although retaining a single cusp dentition (i.e., simple dentition under the authors own definition of dental complexity), besides being mostly carnivorous. It is really important to acknowledge such important exceptions in the text more clearly. They do make mention of snakes at the end of their discussion (lines 257-260), but only to mention that they make other kinds of dental-dietary adaptations (such as the development of venom grooves). However, venom grooves are specific only to viperids, thus representing a small fraction of all crown alethinophidians, and venom grooves are not the criterion of complexity used by the authors nor the morphological variable used in their analyses. Importantly, the pattern in crown alethinophidians and Varanus deviates from the data and results that they actually do get: that an increase in dental complexity does not explain the major taxonomic radiation of these two lineages. I would highly suggest the authors to be more straightforward and explicit on this result, as this does not diminish the impact of their results and makes it clear how unique snakes are in terms of macroevolutionary dynamics relative to other groups of squamates.*

We agree with the reviewer that several clades of unicuspid predatory squamates (namely Mosasauria, crown Alethinophidia, *Chilabothrus*, and *Varanus*) differ from the main pattern we describe for Squamata as a whole and for several individual squamate clades of increased speciation linked to increased tooth complexity and/or plant consumption. We now point to this deviation in the revised Results (lines 172-174) and further develop on these unicuspid clades in the revised Discussion (lines 265-272).

MINOR COMMENTS

- *Line 63: Squamates first appearance in the fossil record is in the Triassic. The estimated origin of the group is in the Permian. The current sentence is ambiguous, so I would suggest making reference here to the estimated divergence time for squamates (not appearance).*

Done. The revised text now refers to the origin of squamates rather than their appearance (line 63).

- *Lines 158-164. The authors make mention of five groups in which speciation increases coincide exactly with increases in tooth complexity or plant consumption: Egeriinae, Polyglyphanodontia, Podarcis, total group Pleurodonta, and a sub-clade of Iguanidae (listed in supplementary table 12). However, Egeriinae (node B) and Podarcis (node D), listed above, do not have any transition in cusp number and the data transition is from insectivorous to omnivorous. This may sound picky but most readers when reading increasing plant consumption may assume transition to herbivory. The text could be improved by stating something like "...or increase in plant consumption (transitions into herbivory or from insectivory to omnivory)."*

Done. We reformulated the sentence to clarify whether these clades showed an increase in tooth complexity, plant consumption, or both, and explicitly stated that Egeriinae and *Podarcis* evolved an omnivorous diet (lines 161-164 of the revised manuscript).

- *Lines 158-164. The authors have enough room to include the names of the five clades (one node above or below) that also face increases in speciation rates along with tooth complexity/diet. It would be much easier for the reader to have those names here, as the authors did for the other five clades, as most will not have the time or patience to look for them on the supplementary tables. The same with the five lineages that faced the opposite pattern (increases in speciation concomitant with decreases in tooth complexity/diet).*

Done. We added the names of all these clades to the revised manuscript (lines 161-170).

- *Lines 170- 171: the authors dropped a comparison between the apex of speciation in squamates with “a critical timing angiosperm evolution” for no apparent reason or connectivity with the rest of the text here. I see the bring some discussion on the evolution of angiosperms and squamates later in the discussion, but the present sentence is out of context. Either rephrase this sentence or simply remove and keep this for the discussion only.*

Done. We removed the reference to angiosperm evolution from this sentence in the revised Results (line 175) and mention this point only in the Discussion.

- *Lines 176-180. I would suggest rephrasing the result to reflect that this explains the evolution of several major components of squamate diversity, but not all of them, most notoriously, crown alethinophidians (one third of all squamate diversity). See other comments above. So authors could write something such as “contributed considerably to the diversification of non-ophidian squamates”.*

Done. In the revised text we now explicitly mention that these traits were key to the diversification of non-ophidian squamates (lines 181-183). Note that diversification rates inferred with hisse were modelled based on our whole super-tree, which is why the resulting model is to be interpreted first for the whole focal group.

- *Discussion: I suggest the authors should add to their discussion the following lines they provided in the response document, as additional factors contributing to diversification dynamics in squamates: “Factors previously identified as influential for squamate diversification include global temperatures (Garcia-Porta et al. 2019; Condamine et al. 2019), latitude (Pyron 2014), and habitat (Ricklefs et al. 2007, Bars-Clozel et al. 2017).”*

Done. We added a sentence referencing these studies to our revised Discussion (lines 233-235).

- *Supplementary Figure 11 | Squamate diversity through time: It’s important to report whether this line were present mean or median estimates across time bins. I find it important to also include the credibility interval around the summary trendline depicted here. However, if you provide net diversification curves (speciation – extinction) from the BAMM results I think it could replace this figure altogether.*

In Supplementary Fig. 11, we display the non-binned, absolute number of lineages through time based on our dichotomous super-tree alone. Therefore, there are no confidence intervals to be

calculated, contrary to estimates based on several trees or inferred from diversification rates. The revised caption more explicitly describes the plot and its origin. We have also added a net diversification curve to Supplementary Fig. 10. We chose to present both items, as Supplementary Figure 11 shows absolute raw diversity from the super-tree, while Supplementary Figure 10 displays the Bayesian rate estimates inferred by BAMM from the tree data, making the two figures complementary.

- *Supplementary table 12. You list all the clades for each node number but would be helpful to repeat here in this legend the five clades with coincident increases in tooth complexity or plant consumption that you mention in the main text. You did this for supplementary table 13, which was helpful, and both for consistency and readership it would be informative here as well.*

Done. We now list all these clades in the revised caption.

Replies to Reviewer #3:

As previously stated, I feel the manuscript was a strong candidate for inclusion in Current Biology as it stood but felt there were places it could better communicate and explore the results. The plots of tooth complexity over time do help the reader visualize the results more readily. I'm happy to recommend acceptance.

We thank the reviewer for their recommendation and the comments they provided, which greatly helped improving the quality of the manuscript.

References

- Alamillo, H. Testing macroevolutionary hypotheses: diversification and phylogenetic implications. (Washington State University, Pullman, 2010).
- Bardet, N., Suberbiola, X., Iarochène, M., Amalik, M., & Bouya, B. Durophagous Mosasauridae (Squamata) from the Upper Cretaceous phosphates of Morocco, with description of a new species of *Globidens*. *Neth. J. Geosci.* **84**, 167-175 (2005).
- D'Amore, D. C. & Blumenschine, R. J. Komodo monitor (*Varanus komodoensis*) feeding behavior and dental function reflected through tooth marks on bone surfaces, and the application to ziphodont paleobiology. *Paleobiology* **35**, 525-552 (2009).
- Fry, B. G. *et al.* A central role for venom in predation by *Varanus komodoensis* (Komodo Dragon) and the extinct giant *Varanus (Megalania) priscus*. *Proc. Natl Acad. Sci. USA* **106**, 8969-8974 (2009).
- Hornung, J. J. & Reich M. Tylosaurine mosasaurs (Squamata) from the Late Cretaceous of northern Germany. *Neth. J. Geosci.* **94**, 55-71 (2015).

- Longrich, N. R., Bardet, N., Schulp, A. S., Jalil, N.-E. *Xenodens calminechari* gen. et sp. nov., a bizarre mosasaurid (Mosasauridae, Squamata) with shark-like cutting teeth from the upper Maastrichtian of Morocco, North Africa. *Cretac. Res.* **123**, 104764 <https://doi.org/10.1016/j.cretres.2021.104764> (2021).
- Losos, J. B. & Greene, H. W. Ecological and evolutionary implications of diet in monitor lizards. *Biol. J. Linn. Soc.* **35**, 379-407 (1988).
- Massare, J. D. Tooth morphology and prey preference of Mesozoic marine reptiles. *J. Vertebr. Paleontol.* **7**, 121-137 (1987).
- Palci, A. *et al.* Plicidentine and the repeated origin of snake venom fangs. *Proc. R. Soc. B* **288**, 20211391 <https://doi.org/10.1098/rspb.2021.1391> (2021).
- Savitzky, A. H. The role of venom delivery strategies in snake evolution. *Evolution* **34**, 1194-1204 (1980).
- Westeen, E. P., Durso, A. M., Grundler, M. C., Rabosky, D. L., & Rabosky, A. R. D. What makes a fang? Phylogenetic and ecological controls on tooth evolution in rear-fanged snakes. *BMC Evol. Biol.* **20**, 80 <https://doi.org/10.1186/s12862-020-01645-0> (2020).

REVIEWERS' COMMENTS

Reviewer #1 (Remarks to the Author):

The authors have successfully addressed all of my comments and I believe the manuscript is ready to be fully accepted for publication. Once again I congratulate the authors on their original research idea and initiative.